# CharLuMA: Efficient Multi-Language Chart-to-Code Generation with Low-Rank Subspace Adaptation

## Abstract

Chart-to-code generation involves translating a chart image into an executable plotting script. However, prior work has largely focused on Python-only solutions, limiting real-world applicability and leaving the learning signals inherent in cross-language equivalences untapped. We argue that aligned multi-language scripts serve as complementary "views" of the same chart, providing mutual guidance to regularize the visual-to-code mapping. As an instantiation of this idea, we introduce CharLuMA – a multimodal large language model (MLLM) that integrates a language-guided mixture of low-rank subspaces into its multimodal projector. This architecture enables parameter-efficient adaptation via dynamic routing to language-specific subspaces, while preserving shared visual-semantic representations of charts. To facilitate training and evaluation at scale, we present Chart2NCode, a dataset of 176k Chart–Python–R–LaTeX quadruples that maintain consistent visual equivalence across languages. Experiments on multiple benchmarks demonstrate that CharLuMA achieves state-of-the-art performance among open-source MLLMs and even surpasses some proprietary systems. Critically, training with more diverse and balanced language sets yields consistent and substantial improvements across all languages by leveraging the rich supervisory signals embedded in cross-language equivalences. Subspace activation analysis further reveals a hybrid allocation pattern, with compact shared cores complemented by broader language-specific zones, while stronger models exhibit smoother and more balanced allocations. Taken together, these results establish multi-language alignment as an effective supervision paradigm for achieving universal chart-to-code generation[1].

## 1 Introduction

Chart-to-code generation is the task of translating charts into executable plotting scripts that accurately reconstruct the underlying data and visual design, positioned at the intersection of visual understanding, code generation, and cross-modal reasoning (Shi et al., 2025). The demand for automated chart reproduction is increasing in various domains such as science, finance, and biology. Recent advances in multimodal large language models (MLLMs) have demonstrated impressive performance across a wide range of vision–language tasks, even approaching human-level capability (Yue et al., 2024; Lu et al., 2023; Wang et al., 2025b; Zhang et al., 2025). Nevertheless, chart-to-code generation remains a particularly demanding problem, requiring models to recover structured data, interpret intricate visual encodings, and produce precise, executable code with strict fidelity.

Existing works focus on translating charts into single-language codes, predominantly using matplotlib in Python (Shi et al., 2025; Zhao et al., 2025; Wu et al., 2025; Belouadi et al., 2024b;a). For example, ChartMimic (Shi et al., 2025) introduced a benchmark with human-curated matplotlib scripts for chart reconstruction, and ChartCoder (Zhao et al., 2025) trained a code-focused large language model (LLM) on large-scale chart–Python pairs. While effective within the Python ecosystem, this line of research overlooks the diversity of plotting libraries and languages used in practice—analysts in many fields rely on R (ggplot2) or LaTeX (TikZ), among others, to create charts.

---

[1]Codes and data are available at `https://anonymous.4open.science/r/CharLuMA-226D`.

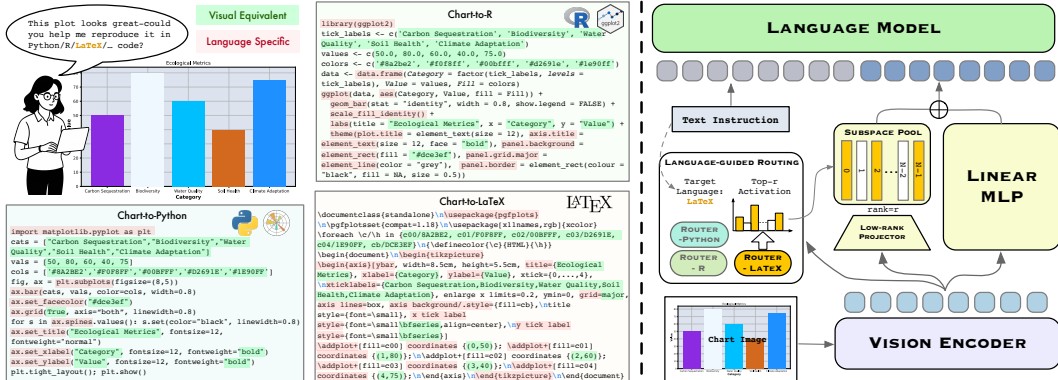

Figure 1: Overview of chart-to-code generation task and CharLuMA architecture. We introduce CharLuMA, a multimodal large language model for chart-to-code generation that augments the multimodal projector with a language-guided mixture of low-rank subspaces.

This single-language focus limits real-world applicability and overlooks a key opportunity: *cross-language alignment*. If a chart is faithfully expressed in different programming languages, those parallel code snippets essentially offer different "views" of the same visualization. Prior works have left the rich learning signal from such cross-language equivalence untapped.

To address these limitations, we introduce **CharLuMA**, a MLLM for chart-to-code generation that is explicitly trained across multiple programming languages. CharLuMA extends a LLaVA-style architecture (Liu et al., 2023) by enhancing its multimodal projector with a language-guided mixture of low-rank subspaces (Figure 1). In essence, we equip the multimodal projector with a low-rank adapter composed of lightweight subspaces, together with a routing mechanism that activates the appropriate combination according to the target language. This design enables parameter-efficient cross-language adaptation: the model learns to share core visual–semantic representations of charts while dynamically adjusting its internal representation to the syntax and conventions of each specific language. By routing through language-specialized subspaces, CharLuMA strikes a balance between cross-language generality and fine-grained language-specific fidelity in code generation. Moreover, training and evaluating a multi-language chart-to-code model requires data that are precisely aligned across languages. Hence, we present **Chart2NCode**, a large-scale dataset of chart images paired with visually equivalent plotting scripts in Python, R, and LaTeX. Chart2NCode contains 176k examples, including a 1k test set, with each example provided as a quadruple: one chart image and three parallel code scripts that faithfully render the same visualization. By enforcing visual consistency across languages, these aligned quadruples offer a rich supervisory signal for learning robust cross-language chart representations.

We demonstrate the effectiveness of our approach through extensive experiments on multiple chart-to-code benchmarks, where CharLuMA achieves state-of-the-art results among open-source MLLMs and even surpasses Claude-Haiku-3.5 and GPT-4o-mini on most metrics. Notably, training with more diverse language sets consistently improves chart-to-code performance, as shown in Figure 4, not only by enhancing cross-language generalization but also by strengthening in-language robustness. When training data are imbalanced across languages, however, the model develops a clear bias toward dominant languages, which limits its universality. This underscores that the learning signals inherent in cross-language equivalences are essential for chart-to-code generation. Subspace activation analysis further shows that CharLuMA allocates subspace capacity in a hybrid manner, with compact clusters shared across languages and broader regions devoted to language-specific specialization, while stronger models exhibit smoother and more balanced allocations.

In summary, our work establishes multi-language alignment as a powerful new paradigm for chart-to-code generation. CharLuMA, together with the Chart2NCode dataset, represents a significant step toward more ***universal and adaptable chart-to-code models***. By leveraging complementary views of the same chart across languages, we show that it is possible to regularize the visual-to-code mapping and achieve more robust, accurate results than ever before. These contributions open the door to chart-to-code systems that can serve a broader range of users and software ecosystems, moving beyond Python-only solutions toward truly language-flexible chart generation.

## 2 RELATED WORK

**Multimodal Large Language Models.** MLLMs employ multimodal projectors to bridge vision encoders with large language models, enabling reasoning across modalities. Models such as BLIP-2 (Li et al., 2022), Flamingo (Alayrac et al., 2022), mPLUG-Owl (Ye et al., 2024), and Qwen-VL (Bai et al., 2023) adopt Q-Formers or resamplers to compress visual tokens for efficient alignment on large-scale image–text corpora. LLaVA (Liu et al., 2023; 2024) extends the instruction-tuning paradigm to the visual domain, demonstrating that a simple MLP projector with one-to-one mapping can effectively align modalities without discarding visual information. Some works (Tong et al., 2024; Lin et al., 2023) explore the combination of various vision encoder to enhance visual representations. More recent work has scaled MLLMs by substituting dense MLP projectors with sparsely gated mixture-of-experts architectures (Xu et al., 2025; Li et al., 2025), which parallelize multiple MLP blocks but incur significant parameter overhead.

**Chart-to-code generation** task requires models to translate chart images into executable plotting scripts, challenging MLLMs with demands in visual understanding, code generation, and cross-modal reasoning. Prior efforts have primarily focused on chart-to-Python generation. Shi et al. (2025) introduced a benchmark of manually curated matplotlib scripts, while Zhao et al. (2025) released a large-scale training corpus. Yang et al. (2024) and Goswami et al. (2025) incorporate user instructions and agent-based methods to enhenace the faithful code synthesis. Other studies untilize chart-to-Python generation for aligning multimodal projectors (Xu et al., 2025) or constructing chart question answering datasets (Zhang et al., 2024b; He et al., 2025). Beyond chart, Belouadi et al. (2024a) and Belouadi et al. (2024b) have developed datasets for image-to-LaTeX generation towards vector graphics. Nevertheless, these efforts remain restricted to single-language settings, which limits practical applicability and overlooks the learning signals in cross-language equivalences.

## 3 THE CHART2NCODE DATASET

We present Chart2NCode, the first large-scale dataset that aligns chart-code pairs across multiple programming languages. With 176k Chart-Python-R-LaTeX quadruples, Chart2NCode establishes a comprehensive resource for developing and evaluating multi-language chart-to-code models.

### 3.1 AUTOMATIC ANNOTATION

We construct multi-language plotting scripts through an automatic annotation pipeline consisting of metadata extraction, template instantiation, and post-debugging (Figure 2). We start by collecting single-language plotting scripts as the source data. ChartCoder (Zhao et al., 2025) provides large-scale Python plotting scripts, while DaTikZ (Belouadi et al., 2024a) contributes extensive TikZ-based codes of scientific vector graphics in LaTeX, from which we extract only the subset corresponding to charts. We further complement these resources by curating 40k R plotting scripts from online platforms and chart galleries (see Appendix B.1).

**Metadata Extraction.** We extract language-agnostic metadata from single-language plotting scripts at the figure, axis, and object levels. The figure level captures global attributes that determine the overall layout and presentation of the chart. The axis level records structural elements that define the coordinate system and its descriptive properties. The object level encodes graphical primitives together with their visual styles, ensuring precise representation of chart content. Metadata are obtained from plotting objects in each language (e.g., `matplotlib.axes` in Python), while LaTeX scripts are processed via regular-expression parsing. Collectively, these layers yield a comprehensive and lossless description of each chart, enabling faithful reconstruction across multiple languages.

**Template Instantiation.** We synthesize multi-language plotting scripts by identifying and filling language-specific templates based on object-level patterns in the metadata. For instance, a horizontal bar chart is characterized at the object level by rectangles of equal height and varying width, which are organized into a data table and matched to the corresponding templates in different languages. Our library comprises 202 human-curated templates spanning 33 chart subtypes in Python, R, and LaTeX, derived from systematic observations of the source data. Once the appropriate template is identified, it is instantiated with structured metadata such as titles, axis ticks, and data values. We

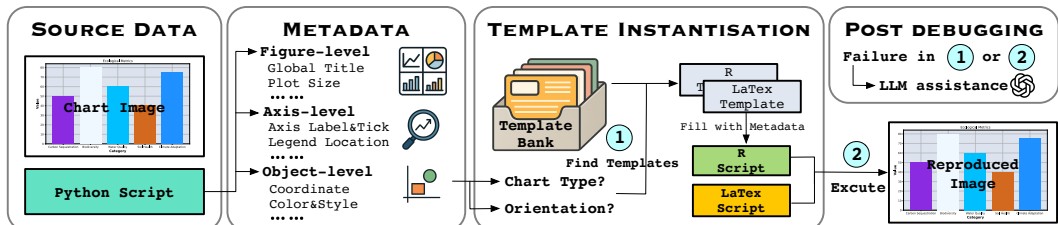

Figure 2: Overview of the automatic annotation pipeline of Chart2NCode.

also add an attribute-mapping process during instantiation to maintain cross-language consistency, such as mapping the `bold` font style in Python to the `bfseries` directive in LaTeX.

**Post Debugging.** In situations where template identifying is unsuccessful or script execution errors occur, we incorporate an LLM-assisted debugging module powered by GPT-4o (OpenAI, 2024b). If no suitable template exists, the module translates the available single-language script into the target languages; if an instantiated template fails, it applies error correction to restore executability. Scripts that remain invalid or produce deprecated figures are discarded to maintain dataset quality.

## 3.2 HUMAN QUALITY CHECKING

We conduct human evaluation to assess the cross-language fidelity of Chart2NCode. A random sample of 1,000 quadruples is independently evaluated by three annotators across four dimensions—structural fidelity, data integrity, semantic consistency, and stylistic coherence—with each dimension rated on a 1–5 scale. The proportion of examples with an average score of at least 4 reaches 97.6% for structural fidelity, 91.6% for data integrity, 95.7% for semantic consistency, and 95.6% for stylistic coherence (see Appendix B.3). These results highlight the robust cross-language consistency of Chart2NCode and its reliability for advancing chart-to-code generation research.

## 3.3 DATA STATISTICS

Chart2NCode encompasses a total of 176k Chart–Python–R–LaTeX quadruples through our automatic pipeline, with 14.7% are refined via LLM-assisted debugging. The dataset spans 15 standard chart types, including bar (18.8%), line (17.1%), scatter (13.2%), radar (5.73%), histogram (4.59%), and box (4.43%). We constructed a test set of 1,000 randomly sampled examples that achieved average scores of at least 4 across all quality aspects in Section 3.2. The average code lengths are 3,998.5 and 4,229.3 characters for the training and test sets, respectively. Comprehensive statistics and details regarding the annotation pipeline are provided in Appendix B.2 and Appendix B.4.

# 4 THE CHARTLUMA MODEL

We propose CharLuMA, a chart-to-code MLLM that extends a LLaVA-style architecture with a novel low-rank subspace adapter for efficient multi-language adaptation. The model is optimized through a progressive training strategy that combines alignment pretraining with instruction tuning.

## 4.1 ARCHITECTURE

CharLuMA is composed of a vision encoder and a LLM backbone, connected through a two-layer MLP projector augmented with a novel low-rank subspace adapter. The adapter is governed by a language-guided routing policy that dynamically selects subspace experts based on both the chart's image features and the target language token, enabling language-specific specialization while maintaining shared visual understanding, as illustrated in Figure 3.

**Vision Encoder.** We adopt SigLIP (Zhai et al., 2023a) as the vision encoder, configured with an input resolution of $384 \times 384$. Pretrained on millions of image–text pairs, it provides strong priors for extracting semantically meaningful visual features. Formally, given a chart input $\mathbf{X}_v$, the vision encoder $g^v(\cdot)$ generates its corresponding representation $\mathbf{Z}_v$, i.e. $\mathbf{Z}_v = g^v(\mathbf{X}_v)$.

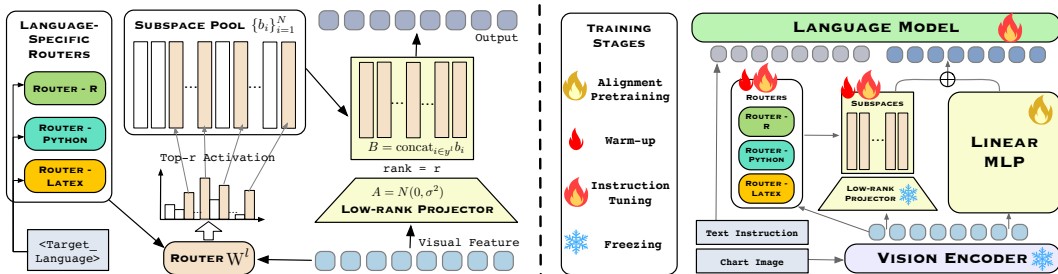

Figure 3: Overview of the CharLuMA architecture and training strategy. The adapter leverages a language-guided routing policy combined with a mixture of low-rank subspaces. The training proceeds through three stages: alignment pretraining, warm-up, and instruction tuning.

**Multimodal Projector.** The standard multimodal projector in LLaVA-style architectures (Liu et al., 2023) is a two-layer MLP block $\mathbf{W}$ that performs a one-to-one transformation, mapping visual features $\mathbf{Z}_v$ into the embedding space of the LLM backbone. The resulting output, $\mathbf{H}_{\text{base}} = \mathbf{W}\mathbf{Z}_v$, serves as a shared base representation across languages.

To enable efficient language adaptation while preserving visual understanding, we augment this linear MLP block with a low-rank subspace adapter (Ding et al., 2025; Wu et al., 2024; Chen et al., 2023). The adapter comprises three components: a low-rank projector $\mathbf{A}$, a language-specific router $\mathbf{W}^l$, and a subspace pool $\{b_i\}_{i=1}^N$. Given the visual features $\mathbf{Z}_v$, the projector $\mathbf{A}$ maps them into a compact rank-$r$ representation ($r < N$). The router then determines which subspaces to activate for the target language $l \in \{\text{Python}, \text{R}, \text{LaTeX}\}$, as specified in the text instruction. Concretely, the router $\mathbf{W}^l$ applies a language-specific transformation to the mean-pooled visual feature $\overline{\mathbf{Z}}_v$, yielding a probability distribution over the subspace pool. The top-$r$ subspaces are then selected, $y^l = \text{top}_r(\text{softmax}(\mathbf{W}^l \overline{\mathbf{Z}}_v))$ where $y^l$ denotes their indices, and concatenated to form the matrix $\mathbf{B} = \text{concat}_{i \in y^l} b_i$. The reconstruction matrix $\mathbf{B}$ is combined with the low-rank projector $\mathbf{A}$ to map the visual features into the LLM embedding space, yielding an language-adaptable representation. The final visual tokens injected into the LLM consists of visual tokens that merge the base and language-adaptable representations:

$$\mathbf{H}_v = \mathbf{H}_{\text{base}} + \mathbf{H}_{\text{adapt}} = \mathbf{W}\mathbf{Z}_v + \mathbf{A}\mathbf{B}\mathbf{Z}_v.$$

**Large Language Model.** We use DeepSeek-Coder (Guo et al., 2024) as the LLM backbone, with 1.3B and 6.7B variants named CharLuMA-1.3B and CharLuMA-6.7B. The visual tokens $\mathbf{H}_v$ produced by the multimodal projector are concatenated with the text tokens $\mathbf{H}_t$ to construct the input sequence for the LLM $g^L(\cdot)$. The final output is then obtained as $g^L(\mathbf{H}_v; \mathbf{H}_t)$.

## 4.2 TRAINING STRATEGY

**Alignment Pretraining.** We initialize the multimodal projector by pretraining the linear MLP block $\mathbf{W}$ on ChartMoE-Align (Xu et al., 2025), a dataset covering 900k Chart–JSON pairs that capture structural elements such as tables, annotations, and styles. The vision encoder and LLM backbone remain frozen during this stage, ensuring that $\mathbf{W}$ learns to align visual features of charts with textual schema representations without altering pretrained components (Yan et al., 2024).

**Instruction Tuning.** We augment the multimodal projector with the proposed low-rank subspace adapter and fine-tune the model on Chart2NCode. We first warm up the language-specific routers $\mathbf{W}^l$ ($l \in \{\text{Python}, \text{R}, \text{LaTeX}\}$) and the subspace pool $\{b_i\}_{i=1}^N$ over fixed steps, while keeping the MLP block, vision encoder, and LLM backbone frozen. The low-rank projector $\mathbf{A}$ is randomly initialized and kept frozen throughout training, ensuring that adaptation capacity is directed toward language-specific diversities rather than redundantly modeling visual commonalities (Ding et al., 2025; Tian et al., 2025). We then unfreeze the LLM backbone and continue training jointly with the routers and subspace pool, while keeping the MLP block, vision encoder, and $\mathbf{A}$ frozen. This progressive protocol stabilizes routing and subspace specialization in the early phase, and subsequently enables the LLM to effectively leverage language-adaptive visual tokens.

## 5 EXPERIMENT

We demonstrate the effectiveness of CharLuMA through comprehensive experiments, which achieve consistent improvements in multi-language chart-to-code generation across diverse benchmarks and surpass competitive baselines.

### 5.1 IMPLEMENTATION DETAILS

During alignment pretraining, we train the MLP block for 1 epoch on 900k Chart–JSON pairs from ChartMoE-Align (Xu et al., 2025), while keeping the vision encoder and LLM frozen, with a learning rate of 2e-4. During instruction tuning, we warm up the subspace pool and language-specific routers for 274 steps, and then continue with full fine-tuning of the LLM backbone for 1 epoch on the Chart2NCode training set, which contains 175k Chart–Python–R–LaTeX quadruples, while keeping the MLP block and vision encoder frozen. The learning rates are set to 2e-4 for the warm-up phase and 2e-5 for fine-tuning. We set the subspace size $N = 32$ and the rank $r = 16$, with detailed analysis provided in Section 6.1. All experiments are conducted with a global batch size of 128 on 8× NVIDIA L40S GPUs. The total training cost is approximately 82 GPU hours for CharLuMA-1.3B and 321 GPU hours for CharLuMA-6.7B. More details are provided in Appendix C.1.

### 5.2 EVALUATION SETTINGS

**Datasets.** We evaluate CharLuMA and the baselines on three chart-to-code datasets. The Chart2NCode test set provides 1,000 charts paired with plotting scripts in Python, R, and LaTeX, enabling multi-language evaluation. ChartMimic (Shi et al., 2025) includes 2,400 charts with human-curated matplotlib scripts in Python, spanning 22 chart types. Plot2Code (Wu et al., 2025) contains 132 high-quality matplotlib plots across 6 plot types.

**Evaluation Metrics.** We assess chart-to-code generation performance from three perspectives: executability, code similarity, and image fidelity. Execution Rate (ER) measures the proportion of generated scripts that run successfully. CrystalBLEU (CB) (Eghbali & Pradel, 2022), a BLEU variant tailored for code, assesses code-level similarity. For image-level fidelity, we adopt DreamSim (DS) (Fu et al., 2023), a fine-tuned metric for perceptual similarity. For Python scripts, we report the averaged F1 score across text, layout, type, and color attributes (Shi et al., 2025), where unexecutable scripts are assigned with zero values. To avoid code similarity inflation for models trained on in-distribution data, we employ image-side GPT-4o scoring (GS) (Shi et al., 2025) on the Chart2NCode test set, where unexecutable scripts are assigned with zero values as well.

### 5.3 BASELINES

**General MLLMs.** We evaluate both closed-source and open-source MLLMs as general-purpose baselines. The closed-source group includes GPT-4o (OpenAI, 2024b), GPT-4o-mini (OpenAI, 2024a), GPT-5-mini (OpenAI, 2025), Claude-3.5-Sonnet (Anthropic, 2024), and Claude-Sonnet-4 (Anthropic, 2025). The open-source group covers representative vision–language models including Qwen3-VL (Team, 2025), InternVL-3.5 (Wang et al., 2025a), GLM-4.5v (Team et al., 2025), DeepSeek-VL (Lu et al., 2024), Phi-3.5-Vision (Abdin et al., 2024), and LLaVA-1.5 (Liu et al., 2023).

**Chart MLLMs.** We also compare against chart-specialized MLLMs tailored for chart reasoning and chart-to-code generation. ChartLlama (Han et al., 2023) extends the LLaVA-v1.5 framework with instruction tuning on multiple chart reasoning tasks. TinyChart (Zhang et al., 2024a) is built on TinyLLaVA (Zhou et al., 2024) for efficient chart understanding. ChartMoE (Xu et al., 2025) advances chart understanding through a mixture-of-experts multimodal projector, integrating chart-to-code generation as a core modality alignment task. ChartCoder (Zhao et al., 2025) directly targets chart-to-code generation by employing a code LLM as its language backbone.

### 5.4 MAIN RESULTS

Existing MLLMs exhibit pronounced disparities in chart-to-code generation across different programming languages, as shown in Table 1. ChartCoder, the state of the art among open-source

Table 1: Performance on ChartMimic, Plot2Code, and Chart2NCode test set. ER ⇑ denotes execution rate, CB ⇑ denotes the code-similarity score CrystalBLEU, DS ⇑ denotes the image-similarity score DreamSim, F1 ⇑ denotes the heuristic F1 score for Python scripts, and GS ⇑ denotes the image-similarity GPT-4o scoring. A "-" indicates that no executable script is generated.

| Models | ChartMimic Chart2Python | | | | Plot2Code Chart2Python | | | | Chart2NCode Chart2Python | | | | Chart2R | | | Chart2LaTeX | | |
|---|---|---|---|---|---|---|---|---|---|---|---|---|---|---|---|---|---|---|
| | ER | CB | DS | F1 | ER | CB | DS | F1 | ER | GS | DS | F1 | ER | GS | DS | ER | GS | DS |
| *Propriety Multimodal Large Language Models* | | | | | | | | | | | | | | | | | | |
| GPT-5-mini | 86.8 | 13.6 | 86.9 | 71.5 | 93.2 | 8.9 | 85.9 | 72.8 | 85.2 | 80.0 | 89.0 | 67.5 | 90.3 | 81.2 | 82.5 | 49.7 | 41.1 | 75.2 |
| GPT-4o-mini | 89.0 | 9.0 | 77.5 | 70.2 | 90.2 | 20.7 | 77.8 | 67.0 | 94.8 | 79.8 | 81.2 | 74.5 | 89.5 | 70.3 | 75.4 | 94.7 | 70.4 | 61.2 |
| GPT-4o | 93.2 | 10.2 | 83.5 | 79.0 | 92.4 | 24.2 | 83.6 | 75.4 | 98.5 | 87.4 | 85.0 | 80.9 | 94.5 | 78.3 | 78.8 | 88.4 | 69.8 | 72.4 |
| Claude-Haiku-3.5 | 88.0 | 7.5 | 76.2 | 65.7 | 87.1 | 16.2 | 72.8 | 56.8 | 91.3 | 76.7 | 81.6 | 68.8 | 93.0 | 73.9 | 76.2 | 78.2 | 55.3 | 57.3 |
| Claude-Sonnet-4 | 96.2 | 13.7 | 83.3 | 79.5 | 95.5 | 12.9 | 81.2 | 76.8 | 98.3 | 88.0 | 86.8 | 81.4 | 93.9 | 83.1 | 82.0 | 92.7 | 72.2 | 76.0 |
| *Open-source Multimodal Large Language Models* | | | | | | | | | | | | | | | | | | |
| Qwen3-VL-2B | 59.0 | 6.2 | 68.9 | 40.4 | 68.9 | 13.0 | 64.2 | 50.1 | 74.0 | 59.6 | 78.0 | 61.0 | 56.5 | 42.0 | 52.4 | 56.0 | 37.4 | 60.8 |
| Qwen3-VL-4B | 78.8 | 7.6 | 71.9 | 59.7 | 77.3 | 12.9 | 66.4 | 55.4 | 87.6 | 77.2 | 83.2 | 76.1 | 75.4 | 60.9 | 66.4 | 62.4 | 45.2 | 68.6 |
| Qwen3-VL-8B | 81.8 | 7.9 | 72.5 | 64.0 | 78.8 | 14.2 | 68.1 | 56.9 | 91.1 | 80.8 | 83.7 | 80.6 | 73.6 | 57.2 | 72.7 | 77.3 | 57.1 | 66.8 |
| InternVL3.5-2B | 51.2 | 4.4 | 67.0 | 32.3 | 61.4 | 12.2 | 55.7 | 44.2 | 69.8 | 53.2 | 76.1 | 53.1 | 61.8 | 44.9 | 53.4 | 9.6 | 4.7 | 52.6 |
| InternVL3.5-4B | 66.6 | 7.7 | 70.1 | 46.0 | 62.1 | 13.3 | 58.8 | 42.7 | 77.9 | 63.4 | 78.4 | 63.0 | 66.8 | 51.5 | 56.4 | 25.7 | 14.7 | 55.5 |
| InternVL3.5-8B | 74.0 | 8.1 | 70.9 | 51.7 | 74.2 | 13.9 | 61.0 | 49.1 | 82.5 | 67.5 | 79.6 | 67.0 | 67.0 | 48.2 | 67.6 | 81.1 | 53.3 | 57.1 |
| DeepSeek-VL-7B | 41.3 | 4.7 | 67.8 | 19.0 | 64.4 | 13.3 | 59.4 | 47.0 | 65.9 | 52.5 | 74.2 | 44.6 | 58.8 | 40.6 | 57.0 | 17.5 | 12.3 | 49.8 |
| Phi-3.5-vision-4B | 66.7 | 6.9 | 44.1 | 38.6 | 72.7 | 14.9 | 63.8 | 42.6 | 68.8 | 56.1 | 53.3 | 34.2 | 47.0 | 33.5 | 52.5 | 7.9 | 5.1 | 42.9 |
| LLaVA-v1.5-7B | 33.0 | 0.7 | 49.6 | 6.7 | 34.9 | 7.1 | 52.1 | 10.4 | 32.9 | 40.2 | 51.9 | 8.9 | 41.4 | 31.0 | 50.7 | 19.4 | 11.7 | 41.0 |
| GLM-4.5v-108B | 88.4 | 8.7 | 73.3 | 67.6 | 83.3 | 13.3 | 80.8 | 56.2 | 85.0 | 79.5 | 85.6 | 77.3 | 85.3 | 70.3 | 77.2 | 80.8 | 63.8 | 62.6 |
| ChartLlama-13B | 70.8 | 0.0 | 45.0 | 15.9 | 81.8 | 4.1 | 50.1 | 22.4 | 65.3 | 14.8 | 46.0 | 16.2 | 13.0 | 6.2 | 44.8 | 81.7 | 49.2 | 32.5 |
| TinyChart-3B | 84.1 | 8.1 | 60.8 | 53.9 | 81.1 | 12.1 | 64.0 | 54.0 | 92.1 | 86.3 | 46.5 | 55.2 | - | - | - | - | - | - |
| ChartMoE-8B | 55.0 | 1.3 | 56.9 | 25.7 | 70.5 | 6.7 | 58.9 | 26.9 | 69.5 | 40.2 | 64.2 | 35.4 | 39.3 | 25.5 | 52.9 | 17.1 | 11.1 | 27.9 |
| ChartCoder-7B | 88.9 | 8.8 | 61.3 | 59.3 | 87.9 | 13.9 | 65.7 | 56.6 | 96.2 | 86.4 | 48.1 | 56.1 | - | - | - | 17.9 | 10.6 | 39.1 |
| CharLuMA-1.3B | 84.8 | 7.3 | 75.1 | 57.5 | 83.3 | 14.5 | 64.3 | 47.2 | 94.4 | 78.4 | 86.5 | 76.9 | 94.5 | 73.3 | 78.9 | 84.5 | 65.1 | 71.3 |
| CharLuMA-6.7B | 91.8 | 8.6 | 79.2 | 70.3 | 96.2 | 15.8 | 68.3 | 60.5 | 98.0 | 88.1 | 88.7 | 83.5 | 96.5 | 80.9 | 81.8 | 89.0 | 74.2 | 72.5 |

systems for chart-to-Python generation, achieves 86.4 GS and 48.1 DS on the Python subset of Chart2NCode, while its performance deteriorates significantly on other languages, with the execution rate dropping to 17.9 on the LaTeX subset and failing to generate valid R scripts. General-purpose open-source models such as DeepSeek-VL-7B and Phi-3.5-Vision show larger imbalances on Chart2NCode, achieving execution rates above 65 on Python but falling below 20 on LaTeX. DeepSeek-VL-7B further exhibits sharp degradation in chart quality, with DreamSim dropping from 74.2 in Python to 57.0 in R and 54.2 in LaTeX. Proprietary models display the same tendency in a more moderate form, as GPT-5-mini and Claude-Haiku-3.5 achieve execution rates above 85 and heuristic F1 scores above 65 on Python, while their performance declines when extended to LaTeX.

CharLuMA effectively addresses the cross-language disparity and establishes itself as the most capable open-source MLLM for general chart-to-code generation. CharLuMA-6.7B delivers the strongest results on well-established chart-to-Python benchmarks among open-source models, achieving 79.2 DS and 70.3 F1 on ChartMimic, and 68.3 DS and 60.5 F1 on Plot2Code. The smaller CharLuMA-1.3B also performs competitively, with 75.1 DS and 57.5 F1 on ChartMimic, and 64.3 DS and 47.2 F1 on Plot2Code, indicating its parameter efficiency. On the multi-language Chart2NCode test set, both models sustain robust and balanced performance across Python, R, and LaTeX. CharLuMA-6.7B achieves 88.7 DS and 83.5 F1 on Python, 81.8 DS and 80.9 GS on R, and 72.5 DS and 74.2 GS on LaTeX, demonstrating consistent generalization beyond Python. Notably, CharLuMA-6.7B outperforms Claude-Haiku-3.5 on most metrics across all benchmarks and delivers performance comparable to GPT-4o-mini on ChartMimic and Chart2NCode. These results underscore CharLuMA's ability to advance open-source chart-to-code generation beyond single-language dominance, narrowing the gap with proprietary systems.

# 6 FURTHER STUDY

We conduct ablation studies and in-depth analyses to disentangle the contributions of different components in CharLuMA, demonstrating its robustness and interpretability.

Table 2: Performance of alternative multimodal projector architectures during the instruction tuning stage of CharLuMA-1.3B and -6.7B on the Chart2NCode test set. Results are averaged over all three languages.

| Model Size | Projector Architecture | Chart2NCode | | |
|---|---|---|---|---|
| | | ER | CB | DS |
| 1.3B | Linear MLP | 88.1 | 14.8 | 76.9 |
| | Mixture-of-MLP | 87.9 | 13.8 | 75.1 |
| | Subspace Adapter | **91.1** | **23.2** | **78.9** |
| 6.7B | Linear MLP | 91.0 | 20.3 | 78.2 |
| | Mixture-of-MLP | 91.9 | 19.3 | 77.4 |
| | Subspace Adapter | **94.5** | **24.5** | **81.0** |

Table 3: Ablation study of subspace settings, router configurations, and training choices in CharLuMA-1.3B on the Chart2NCode test set, with results averaged over all three languages.

| Total Subspace | Activated Subspace | Total Router | Chart2NCode | | |
|---|---|---|---|---|---|
| | | | ER | CB | DS |
| 16 | 8 | 3 | 88.9 | 21.4 | 77.6 |
| 32 | 8 | 3 | 89.4 | 22.1 | 77.8 |
| 64 | 32 | 3 | 87.8 | 19.6 | 75.6 |
| 32 | 16 | 1 | 86.1 | 17.1 | 75.1 |
| 32 | 32 | 0 | 85.8 | 16.6 | 73.2 |
| 32 | 16 | 3 | **91.1** | **23.2** | **78.9** |
| *w/o warming up before finetuning* | | | 87.1 | 18.8 | 75.6 |
| *w/o freezing A matrix of adapter* | | | 90.2 | 21.9 | 78.0 |

## 6.1 MODEL ARCHITECTURE ABLATION

We conduct ablation studies on CharLuMA-1.3B with the Chart2NCode test set to examine alternative architectures, subspace–router configurations, and training choices.

**Alternative Architecture.** We compare our low-rank *subspace adapter* with two alternative projector designs in Table 2. The *linear MLP* block serves as a standard baseline (Belouadi et al., 2024b;a; Zhao et al., 2025) but yields modest improvements, with the 1.3B model staying 88.1 ER and 14.8 CB. The *Mixture-of-MLP* design (Li et al., 2025; Xu et al., 2025) replaces the MLP block with a sparsely gated mixture-of-experts, each initialized from a pretrained MLP block, and we adapt it with a hard-routing policy that always activates the language-specific and shared experts (see Appendix C.2). This raises the execution rate to 91.9 but leads to reduced code and image similarity on the 6.7B model. In contrast, our low-rank *subspace adapter* achieves the strongest results across both model sizes, combining language-aware specialization with parameter efficiency.

**Effect of Subspace Number.** We compare CharLuMA-1.3B under different total and activated subspace settings. In Table 3, rows 1–3 demonstrate that moderate scaling from 16 to 32 subspaces improves diversity and performance, while further expansion to 64 leads to degradation in code accuracy and visual similarity. These results suggest that the 32–16 configuration provides the best balance between expressiveness and efficiency for subspace specialization.

**Effect of Routing Policy.** We compare different routing strategies for activating subspaces in CharLuMA-1.3B. In Table 3, rows 4–5 show that replacing the three language-specific routers with a single shared router reduces CrystalBLEU from 23.2 to 17.1, while removing routers altogether lowers it further to 16.6. These results confirm the importance of language-guided routing policy for maintaining code fidelity and cross-language alignment.

**Effect of Training Choices.** In Table 3, row 7 shows that removing the warming-up stage lowers CrystalBLEU from 23.2 to 18.8 and DreamSim from 78.9 to 75.6, underscoring its role in stabilizing subspace and router specialization. Row 8 shows that unfreezing the $A$ matrix reduces CrystalBLEU to 21.9 and DreamSim to 78.0, indicating that freezing $A$ helps maintain a compact low-rank representation while supporting effective language-specific specialization.

**Effect of Backbone Choices.** We examine the effect of backbone choices in CharLuMA by modifying the language model and vision encoder separately. First, we replace DeepSeek-Coder-6.7B with the general-purpose DeepSeek-LLM-7B while keeping the vision encoder fixed. Second, we replace SigLIP with CLIP-Large with an input resolution of 336 × 336 while retaining the original language model. As reported in Table 4, the default configuration with DeepSeek-Coder-6.7B

Table 4: Ablation study of backbone choices in CharLuMA on the Chart2NCode test set, with results averaged over all three languages.

| Language Model | Vision Encoder | Chart2NCode | | |
|---|---|---|---|---|
| | | ER | CB | DS |
| DeepSeek-LLM-7B | SigLIP | 88.6 | 21.8 | 77.1 |
| DeepSeek-Coder-6.7B | CLIP | 88.8 | 22.0 | 79.2 |
| DeepSeek-Coder-6.7B | SigLIP | **94.5** | **24.5** | **81.0** |

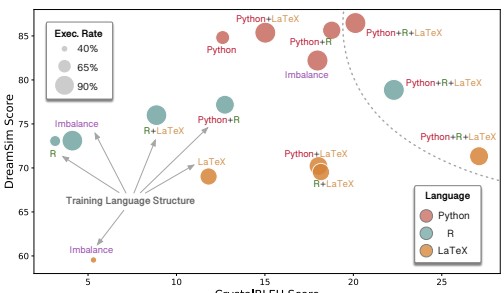 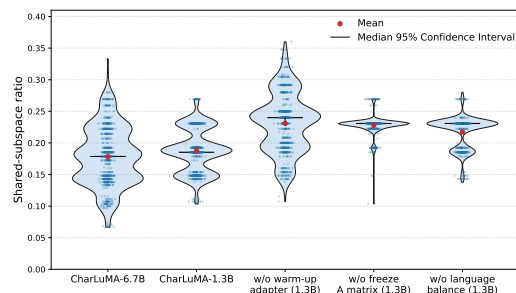

Figure 4: Ablation study of language structure using CharLuMA-1.3B on Chart2NCode.

Figure 5: Distribution of shared-subspace ratios across CharLuMA and ablated models.

and SigLIP achieves the strongest execution rate, CrystalBLEU, and DreamSim scores, whereas each alternative substitution leads to a consistent drop in performance across all metrics.

## 6.2 LANGUAGE STRUCTURE ABLATION

We study the effect of language diversity and balance on the general chart-to-code generation capability of MLLMs by varying the number of programming languages during training, with the number of routers matched to the number of languages. All language configurations are trained with the same strategy and number of steps as CharLuMA-1.3B (see Section 5.1 and Appendix C.2) and evaluated on the Chart2NCode test set restricted to the languages included in training. As shown in Figure 4, greater language diversity leads to substantial improvements. Models trained on three languages achieve the highest execution rates and the strongest code- and image-level similarity scores across all target languages, whereas two-language and single-language settings fall behind by large margins. Moreover, training on the diverse but imbalanced source distribution (76.6% Python, 19.2% R and 4.2% LaTeX) further skews the model toward the dominant language and degrades its performance on the other languages. These results demonstrate that *language diversity enhances both cross-language generalization and in-language robustness by leveraging the learning signals inherent in cross-language equivalences.* At the same time, balanced supervision is critical, as imbalances in the training data introduce systematic biases that undermine universality. Together, these findings underscore the importance of Chart2NCode as the first balanced multi-language dataset for enabling robust and equitable chart-to-code generation.

## 6.3 SUBSPACE ACTIVATION ANALYSIS

We visualize the subspace activation patterns of language-specific routers in CharLuMA-1.3B and CharLuMA-6.7B in Figure 6. The heatmaps display the normalized activation frequency of 32 subspaces for each language and reveal a hybrid allocation of the subspace pool, with compact shared clusters alongside broader language-specific zones. In CharLuMA-1.3B, subspaces 21, 23, and 30 are frequently activated across all languages, while subspace 1 is used primarily for Python, 18 for R, and 17 for LaTeX. In contrast, CharLuMA-6.7B shows a more balanced distribution, with most subspaces—such as 8, 20, and 29—exhibiting intermediate activation frequencies across the three languages, indicating smoother cross-language integration.

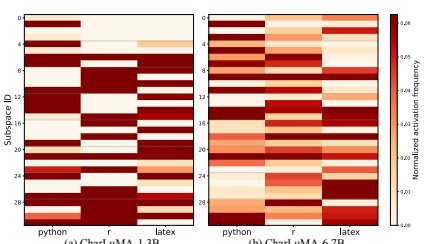

Figure 6: Heatmap of subspace activation frequency for CharLuMA.

We compute the *shared-subspace ratio* to quantify the cross-language allocation of experts. For each chart, it is defined as the proportion of experts simultaneously activated by all language-specific routers relative to the total set of experts activated (see Appendix C.2). Figure 5 reports the distribution of this ratio over a random 1k sample from Chart2NCode. CharLuMA-1.3B achieves a median ratio of 0.19, corresponding to roughly 5 experts shared within a total activation pool of about 27, indicating a compact shared core complemented by broad language-private allocation. CharLuMA-

6.7B shows a similar pattern with a median of 0.18, where about 4.9 experts are shared out of 27.5 on average, suggesting that scaling preserves and slightly reinforces this balanced allocation. In contrast, the ablated 1.3B variants exhibit inflated ratios (0.23–0.24), where more experts are pulled into shared use while the overall activated pool shrinks, indicating weakened specialization.

## 6.4 QUALITATIVE ANALYSIS

We conduct a qualitative analysis that combines error diagnosis of CharLuMA-6.7B and comparisons with GPT-4o and ChartCoder across multiple benchmarks. For error analysis, we find that execution failures often stem from mismatched data dimensions or undefined variables in Python and R, and from syntax issues such as missing braces in LaTeX, while successful runs may still diverge due to missing annotations, misclassified chart subtypes, or stylistic inconsistencies. For model comparison, case studies from Chart2NCode and ChartMimic demonstrate that CharLuMA consistently produces faithful outputs across Python, R, and LaTeX, whereas GPT-4o shows reduced reliability in R and LaTeX, and ChartCoder frequently fails to produce valid code in these two languages. More details are provided in Appendix C.4 and Appendix C.5.

## 7 CONCLUSION

We introduced CharLuMA, a multimodal LLM for chart-to-code generation with a language-guided mixture of low-rank subspaces in its multimodal projector, and Chart2NCode, a dataset of 176k visually aligned Chart–Python–R–LaTeX quadruples. CharLuMA achieves state-of-the-art performance among open-source MLLMs, with ablation studies showing that balanced multi-language training enhances cross-language generalization and mitigates bias toward dominant languages. Subspace analyses further reveal a hybrid allocation of shared and language-specific regions that supports both transfer and fidelity. By leveraging parallel code views of the same chart across languages, we show that cross-language alignment provides a powerful supervisory signal for robust and accurate code generation. These contributions pave the way toward universal, language-flexible chart-to-code systems that better reflect the diverse software ecosystems in practice.

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

## A LLM USAGE

Large Language Models (LLMs) were utilized in this work solely as auxiliary tools for linguistic refinement. Their function was restricted to enhancing grammar, clarity, and stylistic consistency of text that had been originally drafted by the authors. At no stage did LLMs contribute to research ideation, methodological design, data collection, analysis, or interpretation of results. All intellectual contributions, scientific content, and conclusions presented in this paper are entirely attributable to the authors. The authors accept full responsibility for the accuracy, originality, and integrity of the submission, including sections of text that may have been refined with the assistance of LLMs.

## B DATASET

### B.1 DATA ACQUISITION

We collect single-language plotting scripts from established datasets and publicly available repositories as our source data. ChartCoder (Zhao et al., 2025) contributes approximately 160k chart-to-Python scripts, while DaTikZ (Belouadi et al., 2024a) provides 49k vector-graphics-to-Python scripts, of which 8.8k correspond to charts with explicit axis structures. In addition, we curated 40k R plotting scripts from widely used online resources including R gallery [2] and stack overflow [3]. To handle deprecated or non-executable scripts encountered during crawling, we employed GPT-4o as an automated debugging assistant, guided by the prompt instructions in Figure 8, with a total API cost of 132.2 USD.

### B.2 ANNOTATION PIPELINE

**Metadata Structure and Extraction.** We adopt a hierarchical metadata schema to capture chart information at three levels: figure, axis, and object. This structure provides a standardized representation of chart elements across languages while preserving both global properties and fine-grained graphical details. At the figure level, metadata records global properties such as the overall title, background color and legend, plot size (width, height, and units), twin-axis relationships, and subplot layout. For each axis, metadata focuses on type-agnostic attributes including axis titles, x- and y-axis labels, tick values and labels, legends, grids, panel boxes, background color, and annotations. At the object level, metadata captures fine-grained properties of graphical elements grouped into patches, lines, collections, and images. For each object, visual properties such as color, transparency, line width, marker style, and hatch patterns are recorded, together with precise geometric information such as rectangle bounds, circle centers and radii, polygon vertices, line coordinates, scatter offsets, and heatmap arrays. Cleaned labels are associated with color or stylish values where available, ensuring consistency with legends and categorical encodings.

Metadata is extracted by executing or parsing plotting scripts in their native environments. For Python plotting scripts, each script is executed in an isolated runtime, and the figure is inspected using `fig.get_axes()`. Axis-level attributes are gathered through standard APIs such as `ax.get_title()`, `ax.get_xlabel()`, and `ax.get_yticks()`. Object-level elements are obtained by iterating over `ax.patches`, `ax.lines`, `ax.collections` and so on. For R scripts based on `ggplot`, code is evaluated to collect the plotting object p built via `ggplot_build()`. We extract axis-level metadata from structures such as `p$labels$title`, `p$mapping$y`, and `p$theme$panel.border`, while object-level metadata is obtained by iterating over `p$layers`. For base R graphics, we wrap high-level functions like `barplot`, `hist`, and `boxplot`, as well as low-level commands such as `text`, `legend`, and `grid`, to capture

---

[2] https://r-graph-gallery.com/all-graphs.html

[3] Retrieved using StackAPI with keywords representative of R plotting functions and libraries, including `ggplot`, `plot_ly`, `geom`, `plot(`, `hist`, `boxplot` and so on.

---

**Instruction Prompt for Handling Missing Templates in Post-Debugging**

You are provided with a {`original language`} plotting script as shown below. Your task is to transform it to {`target language`} language, starting with "'{`target language symbol`} and ending with "'.
{`original plotting script`}

---

Figure 7: Instruction Prompt for Handling Missing Templates in Post-Debugging

metadata during execution. For LaTeX, we use the regular-expression parsing to detect `axis` environments while drawing commands are parsed to recover object geometries such as rectangles, circles, and paths.

**Template Design.** The templates are parameterized chart skeletons that translate extracted metadata into executable plotting code. Each template specifies placeholders for chart elements such as titles, axis labels, ticks, grids, legends, annotations, and objects, which are directly filled from metadata. The overall structure is consistent across languages, but implementation details differ. Taking the bar type for example, Python uses functions like `ax.bar` or `ax.barh` in matplotlib, R employs `geom_bar` in ggplot, and LaTeX relies on declarative PGFPlots options such as `xbar`, `ybar` and `addplot` using TikZ.

To maintain cross-language consistency during template instantiation, we employ an attribute-mapping process that normalizes visual properties across Python, R, and LaTeX. Legend locations are aligned so that values such as "upper right" in Python correspond to "right" in R and "north east" in LaTeX. Font styles are unified by mapping bold and italic settings into Python's weight and style fields, R's fontface descriptors, or LaTeX commands like `bfseries` and `itshape`. Font sizes are standardized by converting numeric values in Python and R into LaTeX size categories such as `small` or `Large`. Annotation alignment is harmonized by translating Python's top, bottom, and center into equivalent justification values in R and LaTeX. Marker and line styles are also consolidated through shared dictionaries, ensuring that a logical style such as circle, dashed, or cross is rendered consistently across all languages. This mapping guarantees that semantic attributes are preserved even when the syntax differs, allowing metadata extracted in one language to be instantiated in another without loss of fidelity.

**Metadata-Template Matching.** A critical step in our automatic pipeline is to identify the correct template once the metadata of a chart has been extracted. We address this by assigning each chart a type and subtype based on patterns in the object-level metadata. Taking bar charts for example, we examine the geometry of rectangular patches: overlapping intervals reveal stacked bars, repeated clusters of equal size indicate grouped bars, with other cases default to base bars. For pie charts, subtype inference is based on patch geometry and offsets: the presence of an inner radius or nonzero x position signals a donut chart, displaced segment centers indicate exploded pies, and their combination yields donut–exploded pies. These inference rules allow the system to automatically select the most appropriate template across chart variants without manual intervention.

**LLM-assisted Debugging.** We incorporate an LLM-assisted debugging module based on GPT-4o to handle cases where no suitable template can be identified or when an instantiated template fails to execute. Instruction prompts for these two scenarios are provided in Figure 7 and Table 8. The total expenditure on the OpenAI API amounts to 316.6 USD.

Our automatic pipeline finally generates 176K Chart-Python-R-LaTex quadruples, with 14.7% are refined via LLM-assisted debugging. A randomly sampled set of 1,000 examples is reserved as the test set. The average code lengths are 3998.5 and 4229.3 characters for the training and test sets, respectively. The dataset covers a broad range of chart types, including bar (18.8%), line (17.1%), scatter (13.2%), pie (7.3%), ring (5.1%), radar (5.73%), histogram (4.59%), box (4.43%), heatmap (3.56%), violin (3.13%), error point (2.94%), area (2.81%), density (2.79 %), error bar (2.68%), bubble (2.2%), and others (3.64 %).

---

**Instruction Prompt for Failed Template Execution in Post-Debugging**

You are provided with two code snippets. The first is the original code, a {original language} plotting script serving as the reference implementation. The second is the transformed code, a version of the original script translated into {target language}, which is currently unexecutable due to syntax or logic errors.
Original Code: {original plotting script}
Transformed Code: {failed template}
Your task is to identify and correct all errors in the transformed code that prevent it from executing. The corrected script must produce a chart that is semantically equivalent to the one generated by the original code. High-level chart semantics such as axis labels, tick values, bar orientation, or grouping should remain unchanged unless modification is required for successful execution. You may reorder code lines, fix syntax issues, and adjust function arguments as needed. Please output only the corrected code, starting with "'{target language symbol} and ending with "'.

---

Figure 8: Instruction Prompt for Failed Template Execution in Post-Debugging

### B.3 QUALITY ASSURANCE

We conduct a human evaluation to systematically assess the cross-language fidelity of Chart2NCode. We randomly sample 1,000 chart–Python–R–LaTeX quadruples from the Chart2Ncode dataset, which are independently annotated by three annotators. All annotators were recruited on campus, with eligibility requiring prior experience in data visualization and programming in Python, R, and LaTeX. They were compensated in accordance with the institution's standard remuneration policies for academic work. We conduct pairwise evaluations for each quadruple, comparing the reproduced charts in Python, R, and LaTeX against the original image, and annotators assess their fidelity across four dimensions. *Structural fidelity* measures whether the geometric arrangement of the chart is preserved, including the number and configuration of subplots as well as axis orientation. *Data integrity* evaluates whether the underlying quantitative values are reproduced exactly, meaning that the reconstructed chart reflects the same data table as the original. *Semantic consistency* assesses whether textual and categorical information is maintained, ensuring that titles, axis labels, legends, and annotations convey the same meaning without omissions, substitutions, or hallucinations. *Stylistic coherence* concerns the visual presentation, requiring that non-semantic design elements—such as color palettes, font attributes, line styles, gridline visibility, and panel borders—remain consistent with the original chart. All dimensions are rated on a 1–5 scale, where 1 denotes severe mismatch and 5 denotes perfect alignment.

We compute the average per-dimension score across annotators for each example, and report the proportion of examples achieving an average score of at least 4. As shown in Table 5, the evaluation results confirm high fidelity across dimensions: 97.6% of examples exceed the threshold for structural fidelity, 91.6% for data integrity, 95.7% for semantic consistency, and 95.6% for stylistic coherence. To further assess reliability, we compute Fleiss' $\kappa$ on binarized labels (rating $\geq 4$ vs. $< 4$). The resulting average Fleiss' $\kappa$ of 0.83 indicates substantial agreement beyond chance, representing a strong and practical level of consistency for human judgment in chart reproduction tasks.

Table 5: Proportion (%) of examples with average rating $\geq 4$ on 1,000 sampled quadruples, reported per annotator and averaged across annotators. Overall row averages the four dimensions.

| Dimension | Ann. 1 | Ann. 2 | Ann. 3 | Avg. |
|---|---|---|---|---|
| Structural fidelity | 98.3 | 97.1 | 97.5 | 97.6 |
| Data integrity | 90.5 | 91.5 | 92.8 | 91.6 |
| Semantic consistency | 94.9 | 96.6 | 95.7 | 95.7 |
| Stylistic coherence | 96.2 | 95.0 | 95.5 | 95.6 |
| **Overall** | 95.0 | 94.3 | 94.6 | 94.6 |

**Source Script in Python**

```python
import matplotlib.pyplot as plt
import numpy as np

dates = ['2023-10-01', '2023-10-02', '2023-10-03', '2023-10-04', '2023-10-05']
posts = [120, 150, 170, 200, 210]
comments = [60, 80, 90, 100, 110]
shares = [30, 50, 40, 70, 60]

fig, ax = plt.subplots(figsize=(8, 6))

bar_width = 0.2
x = np.arange(len(dates))
palette = ['#FFDAB9', '#191970', '#DEB887']
edge_color = 'black'
bars1 = ax.bar(x - bar_width, posts, width=bar_width, color=palette[0],
edgecolor=edge_color, label='Posts')
bars2 = ax.bar(x, comments, width=bar_width, color=palette[1], edgecolor=edge_color,
label='Comments')
bars3 = ax.bar(x + bar_width, shares, width=bar_width, color=palette[2],
edgecolor=edge_color, label='Shares')

ax.set_title('Social Media Engagement over Days', fontsize=15)
ax.set_xticks(x)
ax.set_xticklabels(dates)
ax.set_ylabel('Count', fontsize=12)
ax.set_xlabel('Date', fontsize=12)
ax.grid(True, which='both', axis='y', linestyle='--', alpha=0.7)
handles, labels = ax.get_legend_handles_labels()
ax.legend(handles, labels, loc='upper right', bbox_to_anchor=(1, 1), ncol=1)
plt.tight_layout()
plt.show()
```

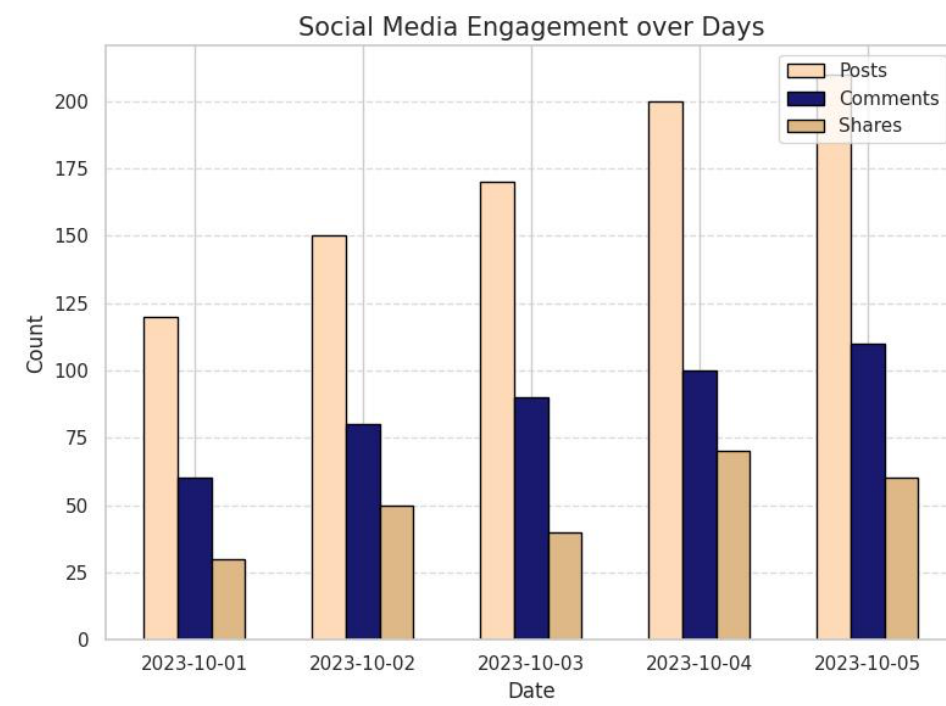

**Metadata**

```
{   "plot_size": {"width": 8.0, "height": 6.0, "unit": "inch"},
    "twin_axes": {},
    "axes_layout": { "n_row": 1, "n_col": 1},
    "facecolor": "#ffffff",
    "ax_0": {
        "type_agnostic": {
            "axis": {"type": "rectilinear", "aspect": "auto"},
            "title": {"content": "Social Media Engagement over Days",
                "size": 15.0, "style": "normal,normal"},
            "x_label": {"content": "Date", "size": 12.0,
                "style": "normal,normal"},
            "y_label": {"content": "Count", "size": 12.0,
                "style": "normal,normal"
            },
            "x_ticks": [{"text": "2023-10-01", "position": ["0",0]},
                {"text": "2023-10-02", "position": ["1",0]},
                {"text": "2023-10-03", "position": ["2",0]},
                {"text": "2023-10-04", "position": ["3",0]},
                {"text": "2023-10-05", "position": ["4",0]}],
            "y_ticks": [{"text": "0", "position": [0,0.0]},
                {"text": “25”, "position": [0,25.0]},
                {"text": “50”, "position": [0,50.0]},
                {"text": “75”, "position": [0,75.0]},
                {"text": “100”, "position": [0,100.0]},
                {"text": “125”, "position": [0,125.0]},
                {"text": “150”, "position": [0,150.0]},
                {"text": “175”, "position": [0,175.0]},
                {"text": “200”, "position": [0,200.0]},
                {"text": “225”, "position": [0,225.0]},],
            "legend": {"exist": true, "loc": 1,"ncol": 1},
            "grid": {"x": true, "y": true},
            "panel_box": true,
            "background_color": "#ffffff",
            "annotation": [],
            "label_to_color": {"Posts": "#ffdab9",
                "Comments": "#191970","Shares": "#deb887"},
            "container_type": [
                "BarContainer","BarContainer","BarContainer"]
        },
```

```
"type_specific": {
    "type": ["bar"], "sub_type": "grouped-bar", "orientation": "vertical",
    "template": ["bar_grouped_vertical_r.jinja", "bar_grouped_vertical_latex.jinja"]}
```

**Metadata-Template Matching**

```
"object": {
"patches": [
{"object_type": "Rectangle", "facecolor": "#ffdab9", "linewidth": 1.0, "linestyle": "solid",
"hatch": null, "geometry": {"x": -0.3, "y": 0.0, "width": 0.2, "height": 120.0}},
{"object_type": "Rectangle", "facecolor": "#ffdab9", "linewidth": 1.0, "linestyle": "solid",
"hatch": null, "geometry": {"x": 0.7, "y": 0.0,"width": 0.2, "height": 150.0}},
{"object_type": "Rectangle", "facecolor": "#ffdab9", "linewidth": 1.0, "linestyle": "solid",
"hatch": null, "geometry": {"x": 1.7, "y": 0.0, "width": 0.2, "height": 170.0}},
{"object_type": "Rectangle", "facecolor": "#ffdab9", "linewidth": 1.0, "linestyle": "solid",
"hatch": null, "geometry": {"x": 2.7, "y": 0.0, "width": 0.2, "height": 200.0}},
{"object_type": "Rectangle", "facecolor": "#ffdab9", "linewidth": 1.0, "linestyle": "solid",
"hatch": null, "geometry": {"x": 3.7, "y": 0.0, "width": 0.2, "height": 210.0}},
{"object_type": "Rectangle", "facecolor": "#191970", "linewidth": 1.0, "linestyle": "solid",
"hatch": null, "geometry": {"x": -0.1, "y": 0.0, "width": 0.2, "height": 60.0}},
{"object_type": "Rectangle", "facecolor": "#191970", "linewidth": 1.0, "linestyle": "solid",
"hatch": null, "geometry": {"x": 0.9, "y": 0.0, "width": 0.2, "height": 80.0}},
{"object_type": "Rectangle", "facecolor": "#191970", "linewidth": 1.0, "linestyle": "solid",
"hatch": null, "geometry": {"x": 1.9, "y": 0.0, "width": 0.2, "height": 90.0}},
{"object_type": "Rectangle", "facecolor": "#191970", "linewidth": 1.0, "linestyle": "solid",
"hatch": null, "geometry": {"x": 2.9, "y": 0.0, "width": 0.2, "height": 100.0}},
{"object_type": "Rectangle", "facecolor": "#191970", "linewidth": 1.0, "linestyle": "solid",
"hatch": null, "geometry": {"x": 3.9, "y": 0.0, "width": 0.2, "height": 110.0}},
{"object_type": "Rectangle", "facecolor": "#deb887", "linewidth": 1.0, "linestyle": "solid",
"hatch": null, "geometry": {"x": 0.1, "y": 0.0, "width": 0.2, "height": 30.0}},
{"object_type": "Rectangle", "facecolor": "#deb887", "linewidth": 1.0, "linestyle": "solid",
"hatch": null, "geometry": {"x": 1.1, "y": 0.0, "width": 0.2, "height": 50.0}},
{"object_type": "Rectangle", "facecolor": "#deb887", "linewidth": 1.0, "linestyle": "solid",
"hatch": null, "geometry": {"x": 2.1, "y": 0.0, "width": 0.2, "height": 40.0}},
{"object_type": "Rectangle", "facecolor": "#deb887", "linewidth": 1.0, "linestyle": "solid",
"hatch": null, "geometry": {"x": 3.1, "y": 0.0, "width": 0.2, "height": 70.0}},
{"object_type": "Rectangle", "facecolor": "#deb887", "linewidth": 1.0, "linestyle": "solid",
"hatch": null, "geometry": {"x": 4.1, "y": 0.0, "width": 0.2, "height": 60.0}}],
"lines": [], "collections": [], "images": []}}}
```

**Script in R**

```r
library(ggplot2)

tick_labels  <- c('2023-10-01','2023-10-02','2023-10-03','2023-10-04','2023-10-05')
group_labels <- c('Posts','Comments','Shares')
values <- list(c(120,150,170,200,210), c( 60, 80, 90,100,110), c( 30, 50, 40, 70,
60))
colors <- c('#ffdab9','#191970','#deb887')
df <- data.frame(
  Category = factor(rep(tick_labels, times = length(group_labels)), levels =
tick_labels),
  Group    = factor(rep(group_labels, each  = length(tick_labels)), levels =
group_labels),
  Value    = unlist(values)
)

p <- ggplot(df, aes(x = Category, y = Value, fill = Group)) +
  geom_bar(stat = "identity", position = position_dodge(width = 0.8), width = 0.8,
show.legend = TRUE) +
  scale_fill_manual(values = colors, breaks = group_labels, labels = group_labels) +
  labs(title = "Social Media Engagement over Days", x = "Date", y = "Count", fill =
"Group") +
  theme(
    plot.title   = element_text(size = 15, hjust = 0.5, face = "plain"),
    axis.title.x = element_text(size = 12, face = "plain"),
    axis.title.y = element_text(size = 12, face = "plain"),
    panel.background = element_rect(fill = "#ffffff"),
    panel.grid.major = element_line(color = "grey"),
    panel.border     = element_rect(colour = "black", fill = NA, size = 0.5),
    legend.position  = "right"
  )
p <- p + scale_y_continuous(breaks = c(0.0, 25.0, 50.0, 75.0, 100.0, 125.0, 150.0,
175.0, 200.0, 225.0), labels = c('0', '25', '50', '75', '100', '125', '150', '175',
'200', '225'))
p <- p + guides(fill = guide_legend(ncol = 1))
print(p)
```

**Script in LaTex**

```latex
\documentclass{standalone}
\usepackage{pgfplots}
\pgfplotsset{compat=1.18}
\usepgfplotslibrary{groupplots}
\usepackage[x11names, rgb]{xcolor}
\definecolor{c00}{HTML}{FFDAB9}
\definecolor{c01}{HTML}{191970}
\definecolor{c02}{HTML}{DEB887}
\definecolor{cb}{HTML}{FFFFFF}

\begin{document}
\begin{tikzpicture}
\begin{axis}[
    ybar, bar width=0.2, width=8.0in, height=6.0in,
    title=Social Media Engagement over Days, title style={font=\large, align=center},
    xlabel=Date, x tick label style={font=\small, align=center},
    ylabel=Count, y tick label style={font=\small, align=center},
    xtick={0, 1, 2, 3, 4}, xticklabels={{2023-10-01}, {2023-10-02}, {2023-10-03},
{2023-10-04}, {2023-10-05}},
    xtick align=center, enlarge x limits=0.2, ymin=0, grid=major, axis lines=box,
    legend style={legend pos=north east, legend columns=1}, axis
background/.style={fill=cb}
]

\addplot+[ ybar, fill=c00, bar shift=-0.180 ] coordinates {
    (0, 120.0) (1, 150.0) (2, 170.0) (3, 200.0) (4, 210.0)};
\addplot+[ ybar, fill=c01, bar shift=0.000 ] coordinates {
    (0, 60.0) (1, 80.0) (2, 90.0) (3, 100.0) (4, 110.0)};
\addplot+[ ybar, fill=c02, bar shift=0.180 ] coordinates {
    (0, 30.0) (1, 50.0) (2, 40.0) (3, 70.0) (4, 60.0)};
\legend{ {Posts}, {Comments}, {Shares} }

\end{axis}
\end{tikzpicture}
\end{document}
```

Figure 9: Case study of annotation pipeline in a vertical grouped bar chart.

## B.4 Case Study

We present two illustrative cases in Figure 9 and Figure 10 to demonstrate the functionality of our annotation pipeline.

Figure 10: Case study of annotation pipeline in a dotted line chart.

## C EXPERIMENTAL SETTINGS AND RESULTS

### C.1 TRAINING AND EVALUATION SETTINGS

We adopt SigLIP (Zhai et al., 2023b) as the vision encoder and DeepSeek-Coder (Guo et al., 2024) as the LLM backbone, yielding two variants of our model: CharLuMA-1.3B and CharLuMA-6.7B. The multimodal connector is implemented as a standard two-layer MLP block augmented with our low-rank subspace adapter.

For alignment pretraining, we train the MLP block for one epoch on 900k chart–JSON pairs from ChartMoE-Align (Xu et al., 2025), while freezing both the vision encoder and LLM, with a learning rate of 2e-4. During instruction tuning, we first warm up the subspace pool and language-specific routers for 274 steps, and then perform full fine-tuning of the LLM backbone for one epoch on 175k chart–Python–R–LaTeX quadruples from Chart2NCode. In this stage, the vision encoder and MLP block remain frozen, the adapter is updated, and the learning rates are set to 2e-4 for warm-up and 2e-5 for fine-tuning. The low-rank projector within the adapter remains frozen throughout. Each training batch is constructed to include all three languages.

All training experiments are conducted with a global batch size of 128 on 8× NVIDIA L40S GPUs. The total training cost for CharLuMA-1.3B is approximately 82 GPU hours, consisting of 35 GPU hours for pretraining, 6 GPU hours for warm-up, and 41 GPU hours for fine-tuning. For CharLuMA-6.7B, the total cost is about 321 GPU hours, including 109 GPU hours for pretraining, 18 GPU hours for warm-up, and 193 GPU hours for fine-tuning. More training hyperparameters are in Table 6.

Table 6: Training hyperparameters for CharLuMA across stages in Section 5.1.

| Hyperparameter | Alignment Pretraining | Warm-up | Instruction Tuning |
|---|---|---|---|
| Learning rate | 2e-4 | 2e-4 | 2e-5 |
| LR schedule | Cosine decay | Cosine decay | Cosine decay |
| Optimizer | AdamW | AdamW | AdamW |
| Max tokens | 2,048 | 2,048 | 2,048 |
| Vision encoder | Frozen | Frozen | Frozen |
| LLM | Frozen | Frozen | Trainable |
| MLP Block | Trainable | Frozen | Frozen |
| Adapter | Frozen | Trainable | Trainable |

For evaluation, we follow a standardized setup across all baselines, fixing the maximum token length to 2,048. The prompting format for the chart-to-code generation task is shown in Figure 11, adapted from Shi et al. (2025). Proprietary MLLMs evaluated include `gpt-4o-2024-08-06`, `gpt-4o-mini-2024-07-18`, `gpt-5-mini-2025-08-07`, `claude-3-5-haiku-20241022`, and `claude-sonnet-4-20250514`, all accessed through their official APIs. For open-source MLLMs, we directly run released checkpoints on NVIDIA L20 GPUs.

### C.2 DETAILED ANALYSIS SETTING

**Alternative Architecture.** We compare our language-guided low-rank subspace adapter with two alternative connector architectures: a linear MLP and a Mixture-of-MLP. In the linear MLP setting, the pretrained MLP block, initialized on chart–JSON pairs, is directly fine-tuned on Chart2NCode. In the Mixture-of-MLP setting, four experts are initialized from the pretrained MLP block, one of which is frozen as a shared expert, while the remaining three serve as language-specific experts. Hard routing is applied such that, in a Python generation task, the Python-specific expert is activated jointly with the shared expert. This setup mirrors the configuration with four experts in total, of which two are activated for each time, as reported in prior studies Li et al. (2025); Xu et al. (2025). Warm-up training is also employed in this setting, followed by continued training with the LLM backbone.

**Language Structure Ablation.** We conduct a language structure ablation to examine the impact of varying the number of plotting languages and corresponding routers during training, while strictly controlling the total number of training steps. In the full three-language configuration, the model is trained on 175k chart images paired with $175 \times 3 = 525k$ plotting scripts, evenly split across

Python, R, and LaTeX. For the single-language setting, we keep the dataset size constant by training on 175k chart–script pairs but increase the number of epochs to three. For the two-language setting, we preserve the same number of training steps by randomly duplicating half of the available plotting scripts to reach the equivalent scale. This ensures that all configurations—one, two, or three languages—are trained under comparable conditions. For the imbalanced configuration, we train on the source data described in Appendix B.1, and maintain the same number of training steps by randomly duplicating samples, as in the two-language setting. The training strategy for all language configurations are the same in Section 4.2.

**Shared Subspace Ratio.** The shared-subspace ratio is a statistic we design to quantify how much different language-specific routers rely on the same experts when processing the same chart. Formally, for each chart example $c$, let $S_{c,l} \subseteq \{0, \ldots, N-1\}$ denote the set of activated experts chosen by the router for language $l$, with $N = 32$ in our standard setting. Each router activates a fixed number of experts (top–$k$, with $k = 16$ in our experiments). Given the set of languages $\mathcal{L}_c$ available for chart $c$, we define $I_c = \bigcap_{l \in \mathcal{L}_c} S_{c,l}$ and $U_c = \bigcup_{l \in \mathcal{L}_c} S_{c,l}$, where $I_c$ is the set of experts shared across all languages and $U_c$ is the total set of experts activated by any language. The *shared-subspace ratio* for chart $c$ is then $R_c = \frac{|I_c|}{|U_c|}$, which lies in $[0, 1]$. A high value indicates that most experts are shared across languages, while a low value indicates that only a few experts are shared and the rest are language-specific.

## C.3 Prompt Sensitivity Study

We adopt the prompt used in ChartMimic (Shi et al., 2025) to maintain experimental consistency, as illustrated in Figure 11. To ensure that the inclusion of the phrase "a STEM paper" does not introduce unintended bias, we conduct a targeted prompt sensitivity analysis. Specifically, we evaluate two variants: (i) removing only the phrase "a STEM paper," and (ii) removing the entire sentence in which it appears. Both ChartLuMA-1.3B and Phi-3.5-vision are assessed on the Chart2NCode test set under these modified prompts. The results in Table 7 indicate that these variations yield no substantive differences in performance, confirming the robustness of our evaluation prompt.

Table 7: Sensitivity study of evaluation prompt on Chart2NCode test set using Phi-3.5-vision and CharLuMA-1.3B.

| Model | Prompt Version | Chart2NCode | | |
|---|---|---|---|---|
| | | ER | CB | DS |
| Phi-3.5-vision | Default | 41.2 | 7.7 | 49.6 |
| | Version 1 | 41.4 | 7.5 | 49.9 |
| | Version 2 | 41.0 | 7.5 | 49.6 |
| CharLuMA-1.3B | Default | 91.1 | 23.2 | 78.9 |
| | Version 1 | 91.0 | 23.1 | 79.1 |
| | Version 2 | 91.2 | 22.9 | 79.0 |

## C.4 Error Analysis

We conduct an error analysis to identify the common sources of execution failures and reproduction limitations of CharLuMA-6.7B. In terms of execution failures, Python and R scripts most frequently break due to mismatched data dimensions or the use of undefined variables, whereas LaTeX scripts typically fail because of syntax omissions, such as missing braces. For example, the Python case in Figure 13(a) produces incompatible x–y list lengths when calling the `ax.plot` function. The R case in Figure 13(b) invokes an undefined variable `angle` in a `geom_polygon` call. The LaTeX case in Figure 13(c) fails due to an omitted closing curly brace in the title and x-tick label definition.

For reproduction limitations, the generated code executes but yields charts that diverge from the reference in various ways. We observe three recurring patterns: (i) annotation errors, such as missing legends or hallucinated axis labels; (ii) chart type errors, where the model misclassifies the intended chart subtype; and (iii) stylistic errors, including incorrect color palettes, font settings, or line styles. For instance, the reproduced chart in Figure 13(a) from ChartMimic mislabels a group name ("AI-Dive" instead of "AIDeepDive") and incorrectly overlays an additional filled area in the radar plot that does not exist in the gold chart. Figure 13(b), also from ChartMimic, shows a subtype recognition error, where stacked error bars are generated in place of grouped error bars. The case in Figure 13(c) from Chart2NCode using R demonstrates malformed x-tick labels (a missing "=") and an ordering of bars inconsistent with the gold chart. Finally, the LaTeX example in Figure 13(d)

---

**Prompt Template of Chart-to-code Generation Task enhanced**

You are an expert {`target language`} developer who specializes in writing matplotlib code based on a given picture. I found a very nice picture in a STEM paper, but there is no corresponding source code available. I need your help to generate the {`target language`} code that can reproduce the picture based on the picture I provide.
Now, please give me the matplotlib code that reproduces the picture below, starting with ""`{target language symbol}`" and ending with """".

---

Figure 11: Prompt Template of Chart-to-code Generation Task

---

**Prompt Template of GPT-4o Scoring enhanced**

You are an excellent judge at evaluating visualization chart plots. The first image (reference image) is created using ground truth matplotlib code, and the second image (AI-generated image) is created using matplotlib code generated by an AI assistant. Your task is to score how well the AI-generated plot matches the ground truth plot.
### Scoring Methodology:
The AI-generated image's score is based on the following criteria, totaling a score out of 100 points: 1. Chart Types (20 points): Does the AI-generated image include all chart types present in the reference image (e.g., line charts, bar charts, etc.)? 2. Layout (10 points): Does the arrangement of subplots in the AI-generated image match the reference image (e.g., number of rows and columns)? 3. Text Content (20 points): Does the AI-generated image include all text from the reference image (e.g., titles, annotations, axis labels), excluding axis tick labels? 4. Data (20 points): How accurately do the data trends in the AI-generated image resemble those in the original image and is the number of data groups the same as in the reference image? 5. Style (20 points): Does the AI-generated image match the original in terms of colors (line colors, fill colors, etc.), marker types (point shapes, line styles, etc.), legends, grids, and other stylistic details? 6. Clarity (10 points): Is the AI-generated image clear and free of overlapping elements?
### Evaluation:
Compare the two images head to head and provide a detailed assessment. Use the following format for your response: — Comments: - Chart Types: ${your comment and subscore} - Layout: ${your comment and subscore} - Text Content: $your comment and subscore - Data: ${your comment and subscore} - Style: ${your comment and subscore} - Clarity: ${your comment and subscore} Score: ${your final score out of 100} — Please use the above format to ensure the evaluation is clear and comprehensive.

---

Figure 12: Prompt Template of Chart-to-code Generation Task

from Chart2NCode exhibits an incorrect color scheme and hallucinates additional text annotations within a pie chart.

## C.5 EXAMPLES

We qualitatively compare CharLuMA-6.7B with GPT-4o and ChartCoder on representative cases drawn from both the Chart2NCode test set and ChartMimic. In the Chart2NCode examples (Figure 15, Figure 16, and Figure 17), CharLuMA-6.7B consistently reproduces high-quality charts across Python, R, and LaTeX, whereas GPT-4o exhibits reduced reliability in R and LaTeX, and ChartCoder frequently fails to generate valid scripts in these languages. We also present four chart-to-Python examples from ChartMimic (Figure 18), which highlight CharLuMA-6.7B's strong chart reproduction ability in Python, performing on par with GPT-4o and ChartCoder, the current state-of-the-art among open-source MLLMs for chart-to-Python generation.

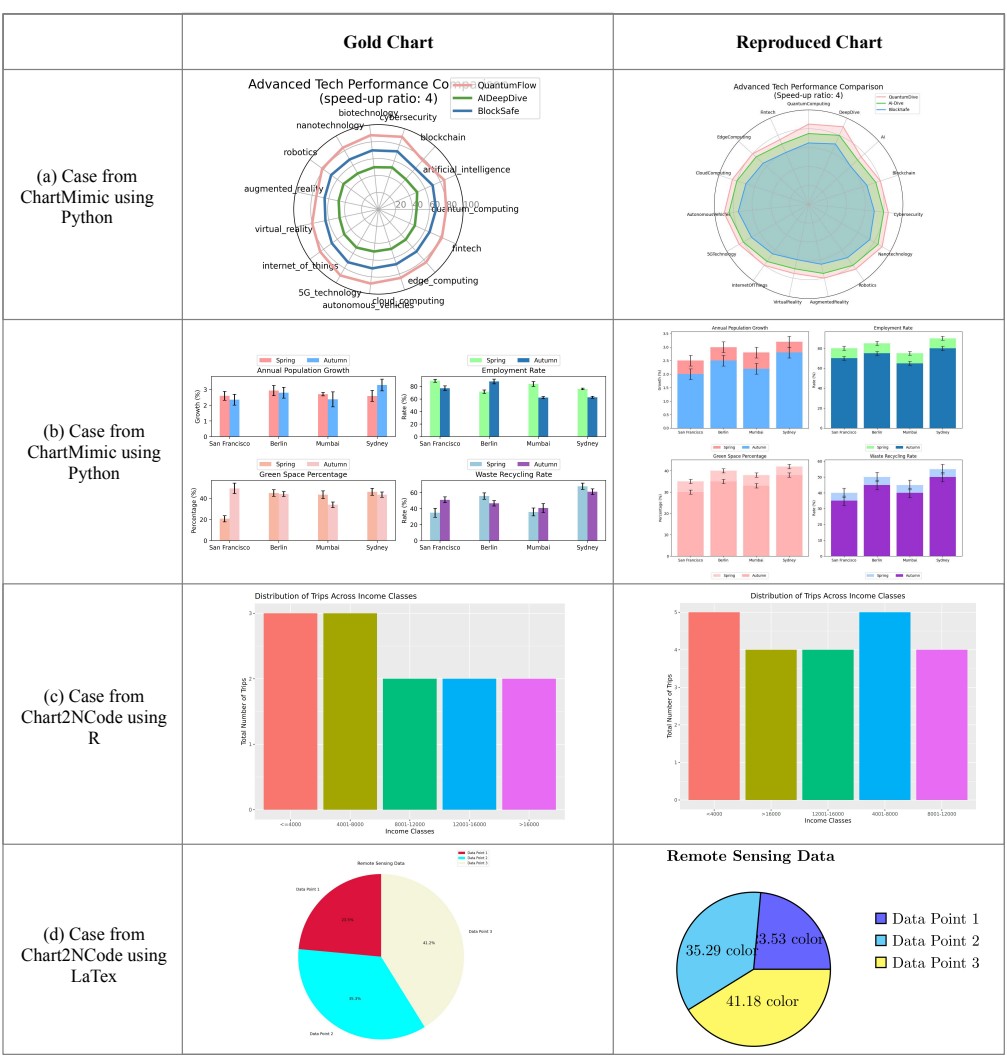

Figure 13: Case study of execution errors in generated code for CharLuMA-6.7B.

Figure 14: Case study of reproduction errors in generated charts for CharLuMA-6.7B.

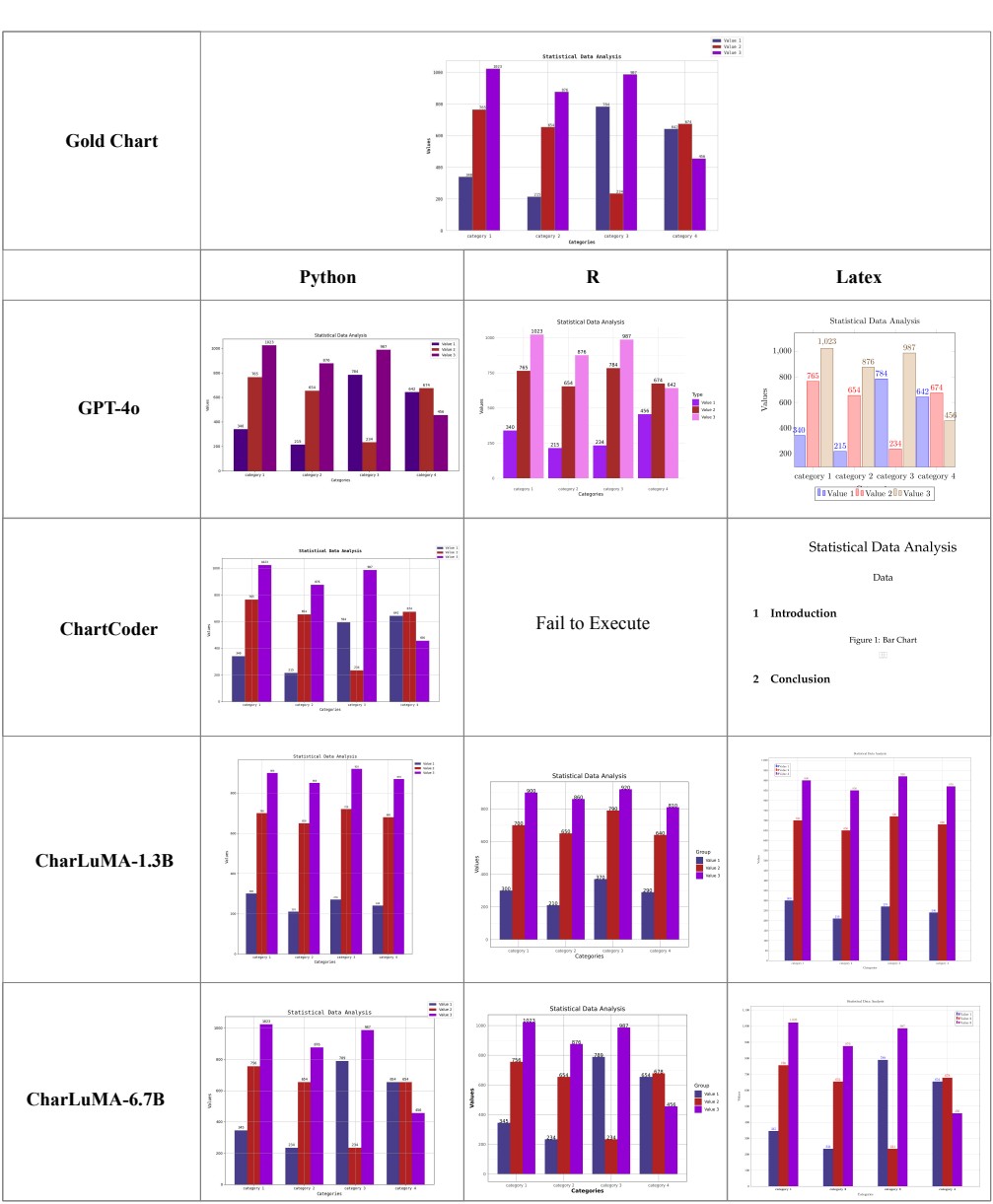

Figure 15: Case study of a grouped bar chart input and generated outputs from the Chart2NCode test set across three plotting languages.

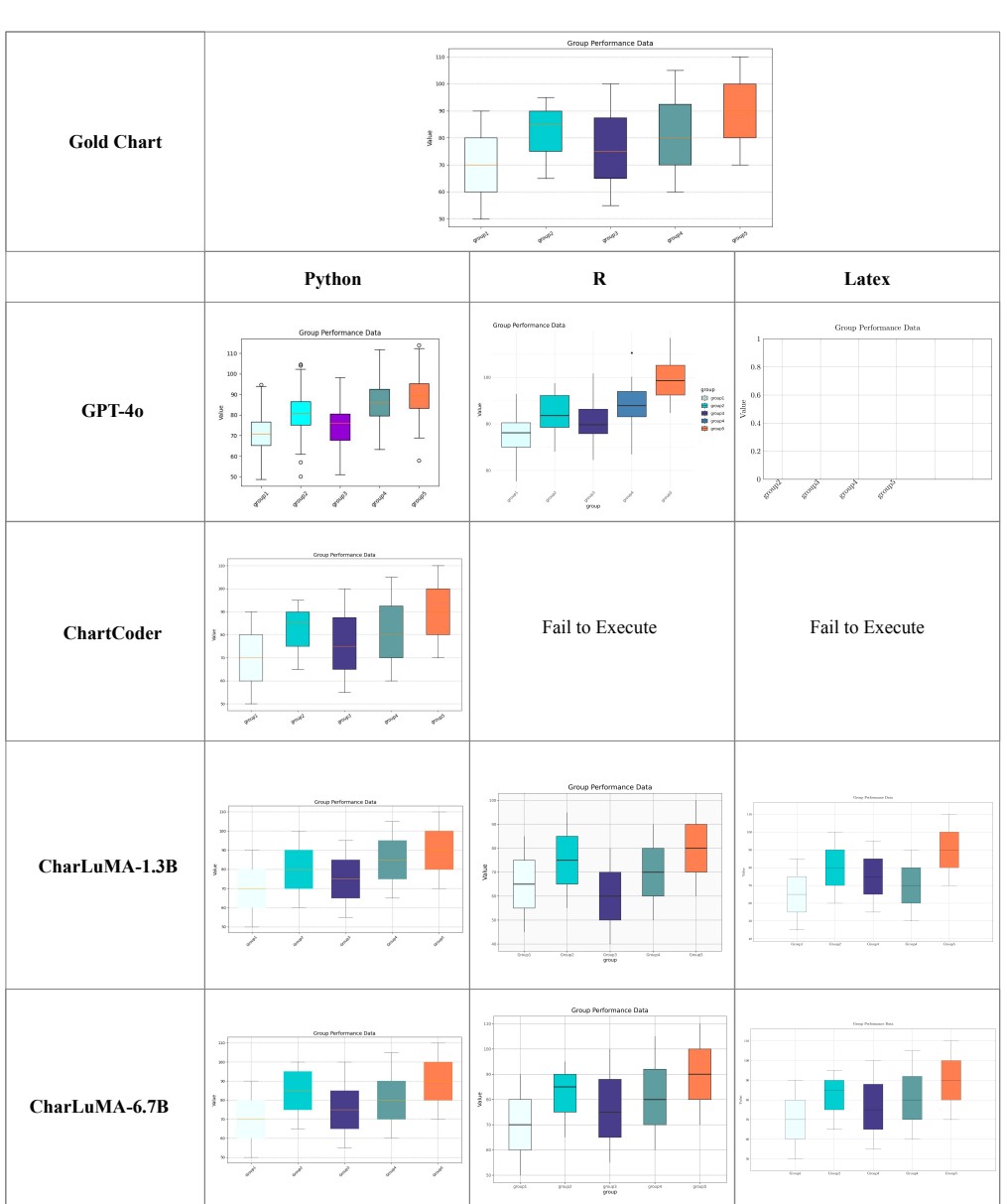

Figure 16: Case study of a box chart input and generated outputs from the Chart2NCode test set across three plotting languages.

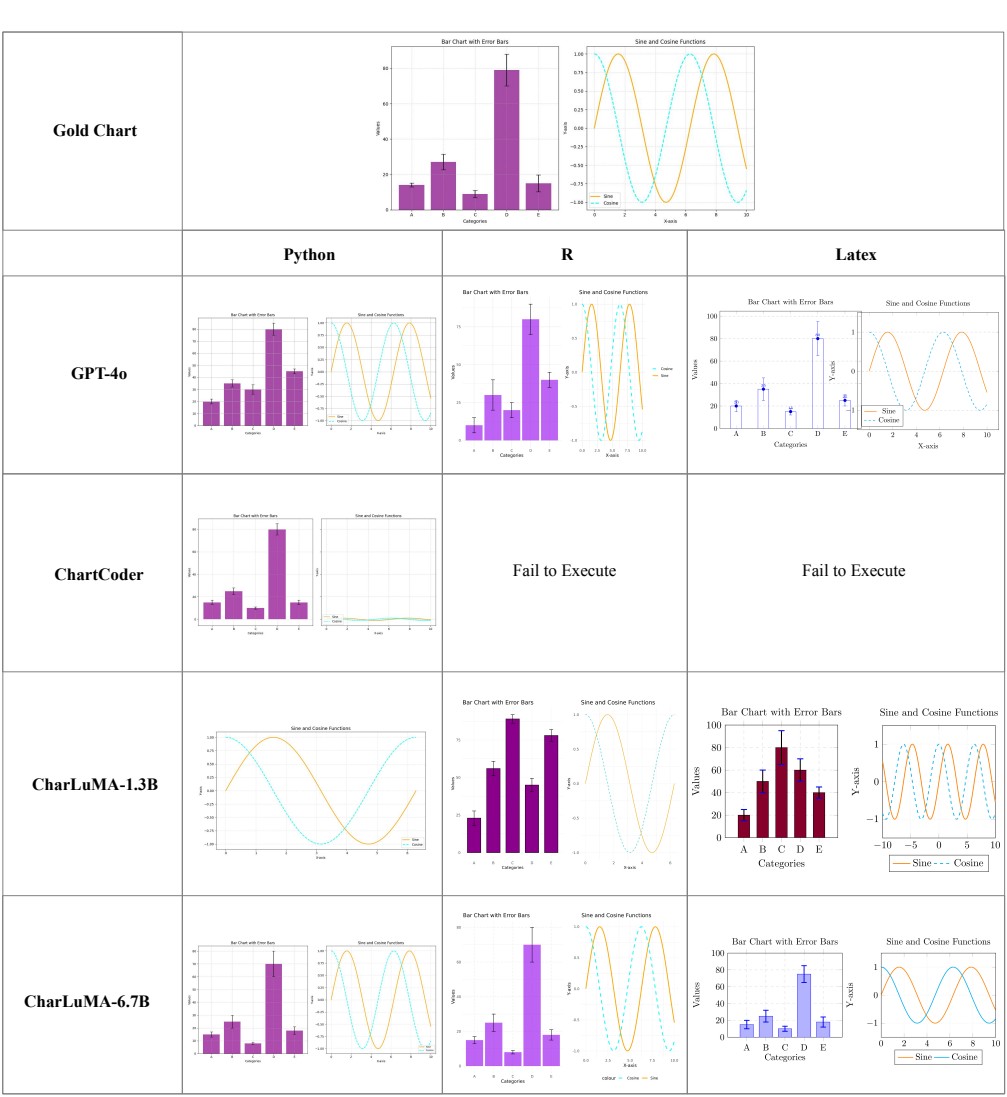

Figure 17: Case study of a two-subplot chart input and generated outputs from the Chart2NCode test set across three plotting languages.

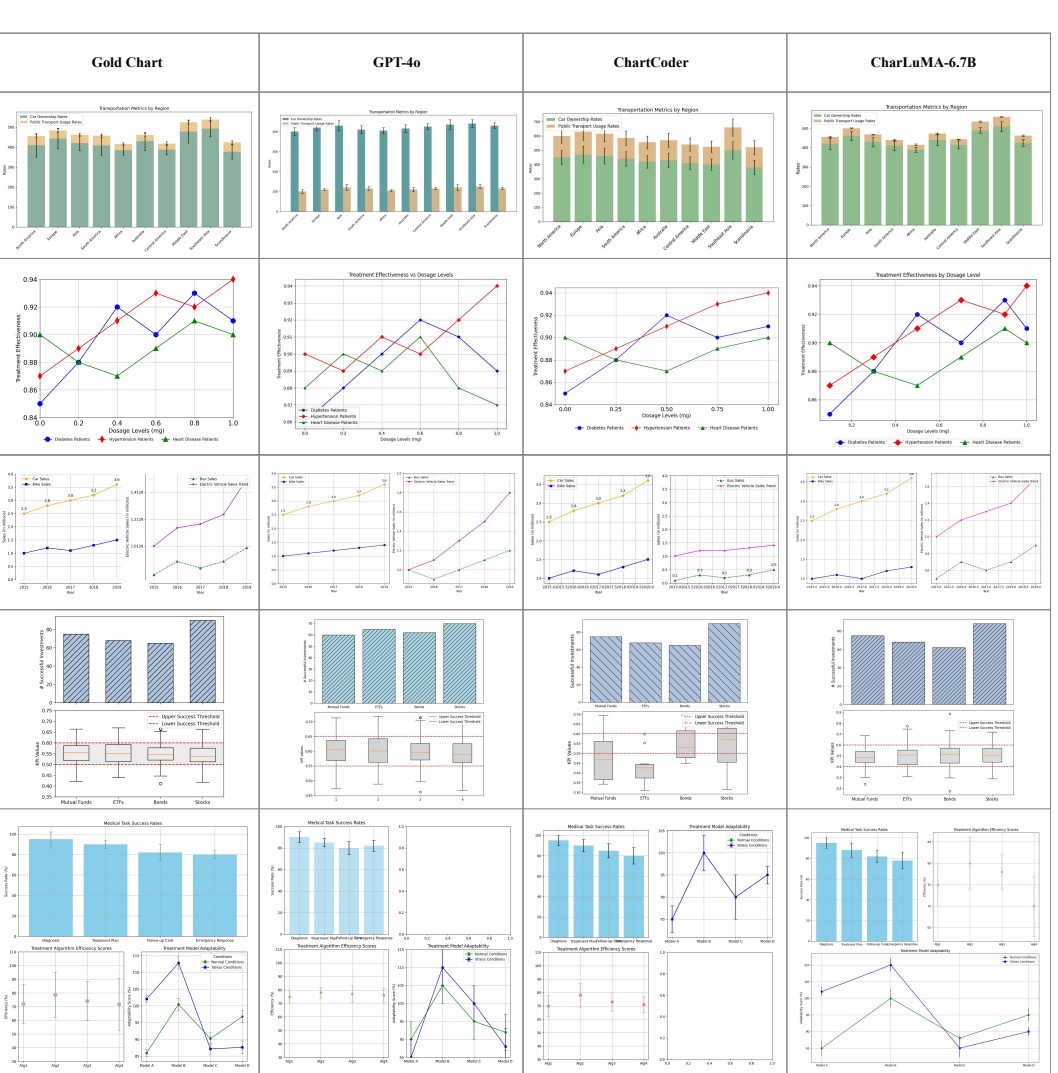

Figure 18: Case study of model inputs and generated outputs from ChartMimic in Python.

