# OpenReview forum: "CharLuMA: Efficient Multi-Language Chart-to-Code Generation with Low-Rank Subspace Adaptation"
_ICLR.cc/2026/Conference — ICLR 2026 Conference Withdrawn Submission_

### Official Review · Reviewer_r3Wy · 2025-10-17

**Soundness:** 2
**Presentation:** 2
**Contribution:** 2
**Rating:** 2
**Confidence:** 4

**Summary:**

The paper presents an auto-generated large-scale multilingual chart-to-code dataset Chart2NCode, and fine-tunes a multimodal LLM based on Deepseek-Coder for chart-to-code generation across multiple languages (Python, R, LaTeX) based on this dataset naming CharLuMA.  The authors claim that CharLuMA achieves state-of-the-art performance among open-source MLLMs and even surpasses some proprietary systems on multiple chart-to-code datasets. The authors also conducts various ablation studies for model choice and subspace activations of different coding languages.

The paper suffers severe disadvantages as the evaluation metrics are mostly similarity metrics and do not include an accuracy metric that evaluates how well the model can solve the problems, and it does not include most recent open-sourced VLMs in evaluations but claims to be SOTA.  I suggest to reject the paper if the authors do not offer sufficient clarifications.

**Strengths:**

1: The paper creates a large multi-lingual chart to code dataset Chart2NCode. The task of generating codes from image-form charts is very helpful at works and highly-valuable.

2: The paper trains a model CharLuMA based on the generated dataset Chart2NCode, the model performs better than a lot of open-source models with similar scales on the chart to code task.

3: The paper conducts comprehensive ablation studies on the model choice and cross-lingual sub-space activation patterns of CharLuMA.

**Weaknesses:**

1: The paper claims that CharLuMA achieves state-of-the-art performance among open-source VLMs, this is not proper given its submission time of 2025.9, as it only compares with open-source VLMs mostly in 2024 and misses more advanced open-source VLM releases like GLM-4.5V and Qwen3-VL.

2: All of the evaluation metrics, ER, CB, DS, F1 and TED, are not real accuracy metrics, they just compare similarities between the generation and reference. None of metrics can tell if the generated chart contains tiny but critical error. The evaluation lacks a true accuracy metric that tells whether the generated chart is fully correct or not.
Moreover, CB, F1 and TED are rewarding codes that are similar to reference codes instead of codes that are correct, and will automatically gives higher score(introduce bias) to models that are trained on Chart2NCode datasets due to learning the unique patterns introduced in the generation process of the Chart2NCode dataset, especially given the dataset is not fully accurate and contains sufficient errors.(See table B.3)

3: The prompt used to evaluate VLMs (shown figure 11) claims the figure is from "a STEM paper", while figures in CharLuMA dataset(as shown in the anonymous link in paper) are mostly in domains of economics and investments, this inaccuracy may cause confusion for other VLMs that are not trained on this pattern.

**Questions:**

1: Is it possible to evaluate based on the real accuracy of the generated figure that whether the figure is fully correct or not instead of only presenting similarity metrics? If this process is hard to be automated you may sample a smaller subset(100 pictures for example) and evaluate them manually via human volunteers and only compare CharLuMA with 2-4 major baselines.

2: What's the performance of more advanced open-source VLMs like GLM-4.5V, or Qwen3-VL? Is CharLuMA still better?

3: According to table B.3, the generation process of Chart2NCode still contains around 5-10% of errors. Have you introduced, or do you plan to introduce any further processes to reduce the errors in the dataset?

---

> ### Author Response · Authors · 2025-11-21
> **Author's Response to Reviewer r3Wy**
>
> _**Weakness 1:** Given the submission time of 2025.9, the paper misses more advanced open-source VLM releases like GLM-4.5V and Qwen3-VL._
>
> Thank you for your advice. Despite that Qwen3-VL models are released on October 15, 2025 after our submission time, We have involved them in our evaluations as shown below. The GLM-4.5V model contains 108B parameters, which exceeds our computational resources. We have updated these baselines accordingly in Section 5.3 and 5.4 of our revised version.
>
> |             | Chart2NCode |             |          |
> |-------------|-------------|-------------|----------|
> | Model       | Exect. Rate | CrystalBLEU | DreamSim |
> | Qwen3-VL-2B | 62.1        | 6.5         | 63.7     |
> | Qwen3-VL-4B | 75.1        | 8.1         | 72.7     |
> | Qwen3-VL-8B | 80.7        | 8.9         | 74.4     |
>
> _**Weakness 2:** All of the evaluation metrics are not real accuracy metrics, they just compare similarities between the generation and reference. Moreover, evaluation metrics in the code side will automatically gives higher score(introduce bias) to models that are trained on Chart2NCode datasets._
>
> The concern likely arises from a misunderstanding of chart-to-code generation task.
>
> First, chart-to-code generation is  **an open-ended task where a single binary correctness metric is not applicable**. Prior works [1,2] use similarity-based metrics that jointly assess code-level and visual fidelity, most of which are included in our evaluation.
>
> Second, the Chart2NCode examples from our automatic pipeline is **composed of visual-grounded content and API templates drawn from official libraries**. This prevents introducing biases related to visual topics or idiosyncratic coding styles. Furthermore, human quality assessment in Appendix B.3 confirms the robust cross-language consistency of Chart2NCode dataset.
>
> Third, CharLuMA-6.7B **delivers the strongest results on two out-of-distribution benchmarks** among open-source models, achieving 79.2 DS and 70.3 F1 on ChartMimic, and 68.3 DS and 60.5 F1 on Plot2Code. Notably, F1 is a code-side evaluation metrics that extracts and evaluates the visual-grounded contents in terms of text, color, layout and type, while ignoring non-visual boilerplate.
>
> _**Weakness 3:** The evaluation prompt in figure 11 claims the figure is from "a STEM paper". This may cause confusion for other VLMs that are not trained on this pattern._
>
> We would like to clarify that **the prompt in Figure 11 follows ChartMimic [2] for experimental consistency**, which includes the phrase “a STEM paper” in the prompt.
>
> To address the reviewer’s concern, we **conducted a prompt sensitivity analysis** by (1) removing only the phrase “a STEM paper,” and (2) removing the entire sentence containing this phrase. We evaluated ChartLuMA-1.3B and Phi-3.5-vision on the Chart2NCode test set. The results below indicate that variations around this phrase lead to no meaningful change in overall performance. We have included this prompt sensitivity analysis in Appendix C.3 of our revised version.
>
> |                |           | Chart2NCode |             |          |
> |----------------|-----------|-------------|-------------|----------|
> | Model          | Prompt    | Exect. Rate | CrystalBLEU | DreamSim |
> | Phi-3.5-vision | Default   | 41.2        | 7.7         | 49.6     |
> |                | Version 1 | 41.4        | 7.5         | 49.9     |
> |                | Version 2 | 41.0        | 7.5         | 49.6     |
> | ChartLuMA-1.3B | Default   | 91.1        | 23.2        | 78.9     |
> |                | Version 1 | 91.0        | 23.1        | 79.1     |
> |                | Version 2 | 91.2        | 22.9        | 79.0     |
>
> [1] Jonas Belouadi, et al. “DETIKZIFY: Synthesizing Graphics Programs for Scientific Figures and Sketches with TikZ.” NeurIPS(2024).
>
> [2] Yang Cheng, et al. “ChartMimic: Evaluating LMM's Cross-Modal Reasoning Capability via Chart-to-Code Generation.” ICLR(2025).

---

> ### Comment · Reviewer_r3Wy · 2025-11-23
> **Thanks for the Clarification**
>
> However, there exists a few issues that the authors need to address.
>
> 1: It seems the authors miss Question 3. Please answer this question.
>
> 2: The authors can use API calls to get access to open-sourced models they cannot run on their device, like GLM-4.5V. There exists a lot of inference providers. https://huggingface.co/zai-org/GLM-4.5V?inference_provider=zai-org
>
> 3: I'm not suggesting Weakness 3 is applied to CharLuMa, as it is specifically trained on this type of data. My point is that weakness 3 may apply to baselines as some baselines may be sensitive to this type of data, especially large propriety models.
> The authors have to ensure all baselines(not only Phi-3.5 vision) are not affected by this prompt flaw. The authors could select 2 or 3 best baselines. However, Phi-3.5V seems not among the best baseline.

---

> > ### Author Response · Authors · 2025-11-30
> > **Further Response to Reviewer r3Wy**
> >
> > **Response 1:** _Have you introduced, or do you plan to introduce any further processes to reduce the errors in the dataset?_
> >
> > Thank you for your reminding. We exclude examples in the test set with average scores below 4 in any aspect mentioned in Section 3.2,  which ensures the robust quality of our evaluation dataset. The noisy examples in the training set are acceptable given the large scale of 176k samples, as models are robust to minor noise during training.
> >
> > **Response 2:** _The authors can use API calls to get access to open-sourced models they cannot run on their device, like GLM-4.5V._
> >
> > Thank you for your clarification. We use the GLM API for z.ai website to run the experiments. The results below show that ChartLuMA-6.7B is still the best performer among open-source models on the chart-to-code generation task.
> >
> > |                         | Chart2NCode |           |          |
> > |-------------------------|-------------|-----------|----------|
> > | Model                   | Exect. Rate | GPT-score | DreamSim |
> > | GLM-4.5V (internvl env) | 83.7        | 71.2      | 75.1     |
> > | CharLuMA-6.7B           | 94.5        | 81.1      | 81.0     |
> >
> > **Response 3:** _Weakness 3 may apply to baselines that are sensitive to this type of data, especially large propriety models._
> >
> > Thank you for your clarification. We have included three more models fro prompt sensitivity study as shown below, including a propriety model claude-sonnet-4 and two leading baselines Qwen3-VL-8B and InternVL3.5-8B. The results below indicate that the variations around the phrase "a STEM paper” lead to no meaningful change in overall performance. Therefore, these models are not sensitive of the phrase  “a STEM paper” in the chart-to-code generation task.
> >
> > |                 |           | Chart2NCode |             |          |
> > |-----------------|-----------|-------------|-------------|----------|
> > | Model           | Prompt    | Exect. Rate | CrystalBLEU | DreamSim |
> > | Claude-Sonnet-4 | Default   | 95.0        | 9.3         | 81.6     |
> > |                 | Version 1 | 94.8        | 9.3         | 81.7     |
> > |                 | Version 2 | 95.1        | 9.1         | 81.8     |
> > | Qwen3-VL-8B     | Default   | 80.7        | 8.9         | 74.4     |
> > |                 | Version 1 | 80.4        | 9.0         | 74.5     |
> > |                 | Version 2 | 80.5        | 9.1         | 74.6     |
> > | InternVL3.5-8B  | Default   | 76.9        | 3.9         | 68.1     |
> > |                 | Version 1 | 76.2        | 3.9         | 67.8     |
> > |                 | Version 2 | 77.0        | 4.1         | 68.4     |

---

### Official Review · Reviewer_4ogQ · 2025-10-27

**Soundness:** 3
**Presentation:** 3
**Contribution:** 2
**Rating:** 4
**Confidence:** 3

**Summary:**

This paper presents CharLuMA, a multimodal large language model for multi-language chart-to-code generation. It introduces a language-guided low-rank subspace adapter that enables efficient and adaptive alignment across Python, R, and LaTeX. The authors also build Chart2NCode, a 176k chart–code dataset supporting balanced multilingual training. Experiments show that CharLuMA achieves state-of-the-art performance among open-source models and narrows the gap with proprietary systems, demonstrating robust cross-language generalization and parameter efficiency.

**Strengths:**

1. The paper introduces Chart2NCode, the first large-scale multi-language chart-to-code dataset covering Python, R, and LaTeX. It fills an essential gap in the community and is constructed through automatic annotation, LLM-assisted debugging, and human validation to ensure data quality and cross-language consistency.

2. CharLuMA achieves state-of-the-art results among open-source models across multiple benchmarks (Chart2NCode, ChartMimic, Plot2Code), demonstrating robust cross-language generalization and even approaching or surpassing closed-source systems like GPT-4o-mini and Claude-Haiku-3.5.

3. The model exhibits excellent parameter efficiency, enabling seamless adaptation across languages without retraining. This makes CharLuMA highly practical and scalable for real-world multimodal and multilingual applications.

**Weaknesses:**

1. Although Figure 5 claims to control for training steps across one-, two-, and three-language settings, the experimental design remains questionable. In the single-language setup, the authors randomly resample and duplicate the same data to match the total number of steps used in the multi-language setting. This repetition increases the model’s exposure to identical samples, potentially causing overfitting and reduced representation diversity. Consequently, the observed improvement under the three-language configuration may partly result from richer and more varied training signals rather than the proposed multilingual routing mechanism itself.

2. The core idea that parallel code snippets serve as complementary "views" to regularize learning is intuitive and effective. However, the paper could be strengthened by explicitly connecting this concept to the extensive body of work in multi-task learning, multi-view learning, and cross-lingual transfer learning. For decades, research in fields like machine translation has shown that training on multiple languages simultaneously can foster more robust and generalized intermediate representations that benefit all constituent tasks.

3. Since the authors have already considered a multilingual setting, in reality there are many more plotting languages beyond the three used in the paper. This training approach can only ensure good performance on those three languages.
If a new language needs to be added, it would still require joint multilingual training to update the subspace structure, rather than being able to adapt effectively through training on the new language alone.

4. The author does not give a base LLM and ViT, nor does he explore the impact of different LLMs.

5. Ablation experiments are inadequate and comparisons may be unfair. The authors used an additional dataset to complete the task, but this dataset may be potentially the most important factor

**Questions:**

see above.

---

> ### Author Response · Authors · 2025-11-21
> **Author's Response to Reviewer 4ogQ**
>
> _**Weakness 1:** Figure 5 claims to control for training steps across one-, two-, and three-language settings. The data repetition in the single-language setup increases the model’s exposure to identical samples, potentially causing overfitting and reduced representation diversity._
>
> Thank you for your comment. We would like to clarify that our language structure ablation **controls the number of training steps and maintains consistent visual inputs across settings** by applying data re-sampling in the one- and two-language configurations.
>
> To further address the reviewer’s concern, we conduct an **additional ablation in which the two- and three-language training sets are downsampled to match the training steps of the one-language setting, despite the resulting reduction in available visual inputs**. Specifically, we sample 2/3 of the quadruples for the two-language setting and 1/3 for the three-language setting, resulting in a loss of one-third and two-thirds of the visual inputs, respectively. **Under this constrained setup, the three-language model remains the top performer** across all evaluation languages, followed by the two-language models, as shown below.
>
> | Training Setting | Evaluation Language | Exect. Rate | CrystalBLEU | DreamSim   |
> |------------------|---------------------|-------------|-------------|------------|
> | Python+R+LaTex   | Python              | 91.9       | 18.1  | 82.2 |
> | Python+R         | Python              | 79.8       | 16.3  | 76.6 |
> | Python+LaTex     | Python              | 91.4       | 13.1  | 79.5 |
> | Python           | Python              | 71.2       | 11.7  | 77.6 |
> | | | | |
> | Python+R+LaTex   | R                   | 89.6       | 20.8  | 74.1 |
> | Python+R         | R                   | 80.7       | 11.7  | 71.7 |
> | R+LaTex          | R                   | 85.6       | 7.7    | 70.0 |
> | R                | R                   | 58.3       | 3.3  | 67.9 |
> | | | | |
> | Python+R+LaTex   | LaTex               | 81.6       | 25.4  | 69.0 |
> | R+LaTex          | LaTex               | 73.2       | 17.4  | 65.5 |
> | Python+LaTex     | LaTex               | 80.8       | 15.7  | 64.4 |
> | LaTex            | LaTex               | 72.7       | 10.3  | 58.7 |
>
> _**Weakness 2:** The paper could be strengthened by explicitly connecting this concept to the extensive body of work in multi-task learning, multi-view learning, and cross-lingual transfer learning._
>
> Thank you for your comment. Our work is indeed **inspired by multi-view learning**, where complementary representations provide rich supervision signals. In our setting, aligned Python, R and LaTex scripts constitute visual-equivalent yet language-specific views of the same chart. To our knowledge, this multi-view or cross-lingual perspective **has not been explored in prior chart-to-code research, and our work serves to bridge this gap**.
>
> Notably, we have **examined the multi-task learning architecture of MLLM in our preliminary experiment**, including ChartMoE [1] and CuMo [2]. These models employ mixtures of parallel experts for handling multiple tasks, whereas plotting scripts in different languages are coupled through shared visual semantics.
>
> Motivated by this, we adopt a shared pool of low-rank subspaces with language-guided routing, which yields optimal performance in this multi-language setting. This design enhances both cross-language generalisation and in-language robustness in our experiments, while the prior multi-task architecture is included as baselines **in our ablation study in Section 6.1**.
>
> _**Weakness 3.** There are many more plotting languages beyond the three used in the paper. If a new language needs to be added, it would still require joint multilingual training to update the subspace structure, rather than being able to adapt effectively through training on the new language alone._
>
> Our architecture is designed to support efficient adaptation across multiple plotting languages, **enabling new languages to be incorporated through lightweight language-specific routers**. Incremental learning approaches such as MAD-X [3] and AdapterFusion [4] demonstrate that additional languages or tasks can be integrated through modular, invertible adapter designs. This paradigm aligns naturally with our language-guided mixture of low-rank subspaces, which also enables modular, language-specific updates. Extending our model with such incremental-adaptation techniques is a promising direction for future work but is beyond the scope of the current paper.
>
> [1] Xu Zhengzhuo, et al. “ChartMoE: Mixture of Diversely Aligned Expert Connector for Chart Understanding.” ICLR(2025).
>
> [2] Li Jiachen, et al. “CuMo: Scaling Multimodal LLM with Co-Upcycled Mixture-of-Experts.” NeurIPS(2024).
>
> [3] Jonas Pfeiffer, et al. “MAD-X: An Adapter-Based Framework for Multi-Task Cross-Lingual Transfer.” EMNLP(2020).
>
> [4] Jonas Pfeiffer, et al. “AdapterFusion: Non-Destructive Task Composition for Transfer Learning.” EACL(2021).

---

> ### Author Response · Authors · 2025-11-21
> **Author's Response to Reviewer 4ogQ  (continue)**
>
> _**Weakness 4:** The author does not give a base LLM and ViT, nor does he explore the impact of different LLMs._
>
> Thank you for your question. We conducted additional ablations to examine the impact of different backbone choices. Specifically, we replaced the language model with DeepSeek-LLM-7B and the vision encoder with CLIP-large-336. The results below show that our default configuration—DeepSeek-Coder-6.7B with SigLIP—remains the strongest combination.
>
> We have incorporated this comparison into Section 6.1 of the revised paper.
>
> |               |                | Chart2NCode |             |          |
> |---------------|----------------|-------------|-------------|----------|
> | LLM Backbone  | Vision Encoder | Exect. Rate | CrystalBLEU | DreamSim |
> | DeepSeek-LLM-7B     | SigLIP         | 88.6        | 21.8        | 77.1     |
> | DeepSeek-Coder-6.7B | CLIP           | 88.8        | 22.0        | 79.2     |
> | DeepSeek-Coder-6.7B | SigLIP         | 94.5        | 24.5        | 81.0     |
>
> _**Weakness 5**: Ablation experiments are inadequate and comparisons may be unfair. The authors used an additional dataset to complete the task, but this dataset may be potentially the most important factor_
>
> We would like to **request clarification on what the reviewer refers to as the “additional dataset.”** If this relates to the language-structure ablation, please see our response to Question 1; otherwise, further clarification would be helpful.
>
> We also emphasize that **our ablation study is extensive** and conducted in a controlled manner, covering alternative architectures, subspace configurations, routing policies, and training choices to ensure fair and meaningful comparisons.

---

### Official Review · Reviewer_3jtc · 2025-10-29

**Soundness:** 3
**Presentation:** 3
**Contribution:** 3
**Rating:** 6
**Confidence:** 4

**Summary:**

This paper introduces **CharLuMA**, a multimodal large language model for **universal chart-to-code generation** across Python, R, and LaTeX. Unlike prior Python-only approaches, CharLuMA leverages **cross-language alignment** to improve visual-to-code mapping through a **language-guided mixture of low-rank subspaces**. The authors also release **Chart2NCode**, a dataset of 176k Chart–Python–R–LaTeX quadruples ensuring visual consistency. Experiments show that CharLuMA achieves **state-of-the-art performance**, demonstrating that multi-language supervision significantly enhances generalization and code quality.

**Strengths:**

* Studies an essential and timely area of cross-lingual learning in the context of chart-to-code generation.
* Proposes a new multimodal architecture (CharLuMA) supporting chart-to-code generation in three languages: Python, R, and LaTeX.
* Demonstrates strong performance, outperforming existing baselines and even some proprietary models.
* Releases models and a large, well-structured dataset (Chart2NCode) containing 176k visually consistent Chart–Code quadruples.
* The paper is well written, clearly presented, and thoroughly executed, with comprehensive experiments and analyses.

**Weaknesses:**

* The contribution novelty is somewhat limited, as it mainly extends LLaVA to a new application rather than introducing a fundamentally new modeling approach.
* In Figure 5, the imbalance setting lacks reported results for the LaTeX language.
* The metadata construction process requires more clarification and illustration. A concise figure (adapted from the appendix) showing how metadata are extracted from source scripts—perhaps with annotations or visual markings—would improve clarity. It is also unclear whether metadata are extracted via execution, parsing, or the use of LLMs. Although the authors mention an automatic pipeline, the released codebase does not appear to include these components.
* The choice of the vision encoder (SigLIP) is not well justified or compared against alternative encoders.
* Only three languages are considered (Python, R, LaTeX), which limits the generality of the “multi-language” claim and raises questions about scalability.

**Questions:**

Q: How do u really get the meta data

---

> ### Author Response · Authors · 2025-11-21
> **Author's Response to Reviewer 3jtc**
>
> _**Weakness 1:** The contribution novelty is somewhat limited, as it mainly extends LLaVA to a new application rather than introducing a fundamentally new modeling approach._
>
> Thank you for the comment. Our contribution extends beyond applying LLaVA to a new task by **introducing a novel multi-language modeling formulation supported by a large-scale aligned dataset**.
>
> We argue that different programming languages serve as complementary “views’’ of the same chart, with shared visual semantics providing mutual guidance for regularizing the visual-to-code mapping.
>
> Architecturally, we propose **a language-guided mixture of low-rank subspaces** in the multimodal projection layer, enabling effective cross-language knowledge sharing while improving robustness within each individual language.
>
> On the dataset side, we construct **a large-scale chart-to-code dataset that aligns Python, R, and LaTeX** plotting scripts into visual-equivalent chart quadruples, providing a unified foundation for systematic multilingual chart-to-code learning.
>
> _**Weakness 2:** In Figure 5, the imbalance setting lacks reported results for the LaTeX language._
>
> The LaTeX result under the imbalanced setting is included in Figure 5. It corresponds to **the point located at the lower-left region of the scatter plot**. In the figure, the text annotation indicates the training configuration, while the dot color denotes the evaluation language, which includes LaTeX.
>
> _**Weakness 3:** The metadata construction process requires more clarification and illustration. It is also unclear whether metadata are extracted via execution, parsing, or the use of LLMs. The released codebase does not appear to include these components._
>
> Thank you for the advice. First, **due to page limits**, the details of automatic annotation pipeline and illustrative examples are placed in **Appendix B.2 and Appendix B.4**. Figures 9 and 10 demonstrate the functionality of the pipeline through case study, including how metadata are extracted from source scripts. We will further refine the overview figure in the main text with additional annotations and visual markings in the next revision of our paper.
>
> Second, **as stated in Lines 152–154**, metadata for Python and R are obtained through controlled execution, whereas LaTeX metadata are extracted via structured parsing, with LLM assistance when necessary.
>
> Third, we have added the relevant components of the automatic pipeline to the anonymous GitHub repository. The implementation can be found under the _dataset\_construction_ directory.
>
> _**Weakness 4**: The choice of the vision encoder (SigLIP) is not well justified or compared against alternative encoders._
>
> Thank you for your comment. SigLIP is chosen as our vision encoder due to its strong image–text alignment and robustness on fine-grained visual details
>
> To further validate this choice, we conducted an additional ablation by using CLIP-large-336 under CharLuMA-6.7B. As shown below, **SigLIP yields consistently better performance than CLIP** across all evaluation metrics on Chart2NCode test set. These results indicate that SigLIP is the more suitable encoder for chart-to-code tasks.
>
> We have incorporated this comparison into Section 6.1 of the revised paper.
>
> |                | Chart2NCode |             |          |
> |----------------|-------------|-------------|----------|
> | Vision Encoder | Exect. Rate | CrystalBLEU | DreamSim |
> | CLIP           | 88.8        | 22.0        | 79.2     |
> | SigLIP         | 94.5        | 24.5        | 81.0     |
>
> _**Weakness 5:** Only three languages are considered (Python, R, LaTeX), which limits the generality of the “multi-language” claim and raises questions about scalability._
>
> We select Python, R and LaTex because the **they are the most widely used languages** for data visualisation and scientific reporting.
>
> According to the Kaggle Machine Learning & Data Science Survey [2], Python, SQL and R remain the three most common programming skills for data scientists, where SQL is not used for visualisation. LaTeX is a primary system for scientific writing, with many publication-quality figures in STEM fields produced directly through LaTeX workflows and supported by major academic publishers.
>
> [1] openai/clip-vit-large-patch14-336 from huggingface
>
> [2] Paul Mooney. 2022 Kaggle Machine Learning & Data Science Survey.

---

> ### Author Response · Authors · 2025-11-21
> **Author's Response to Reviewer 3jtc (continue)**
>
> _**Question 1**: How do u really get the meta data_
>
> Thank you for the question. **The metadata extraction process is described in detail in Section 3.1 and Appendix B.2.** In brief, metadata are obtained by executing or parsing plotting scripts within their native environments.
>
> For Python, each script is executed in an isolated runtime, and the resulting figure is programmatically inspected through _fig.get\_axes()_. For R, metadata are retrieved from the internal ggplot object representation via _ggplot\_build()_. For LaTeX, metadata are obtained through structured parsing: axis environments (e.g., _\begin{axis}_) are detected using regular expressions, and drawing primitives (such as rectangles, circles, and paths) are parsed to recover object geometries.
>
> Illustrative examples of the automatic annotation pipeline are provided in Appendix B.4.

---

### Official Review · Reviewer_A9S1 · 2025-11-01

**Soundness:** 3
**Presentation:** 3
**Contribution:** 2
**Rating:** 4
**Confidence:** 4

**Summary:**

This paper focuses on the chart-to-code field, extending the previous chart-to-Python approach to LaTeX R and Python. The paper presents 176k training data sets and trains charLUMA, a chart-to-code model based on the MOE architecture.

**Strengths:**

1. Chart-to-Ncode is a novel field. Unlike previous work that used only single Python code, extending the task to multiple code types better reflects real-world needs.

2. ChartMULA achieved state-of-the-art (SOTA) results on various benchmarks, demonstrating the effectiveness of the model training strategy and training data.

**Weaknesses:**

1. There appears to be some key information missing, such as the statistical breakdown of different code types in the training set. I am also unsure how the distribution mentioned in Section 6.2 (76.6% Python, 19.2% R, and 4.2% LaTeX) corresponds to the results in Figure 5 and how this ratio was sampled. To my knowledge, the DaTikZ dataset does not contain a large amount of chart-to-LaTeX code, and as stated in Section 3.1, the chart-to-R data is limited to 40k samples at most. Does this imply that the vast majority of the data is chart-to-Python and originates from ChartCoder? If so, the contribution of the proposed dataset might be weakened. My brief review of the images in the anonymous link seems to support this assumption, as most of the data appears to have been rendered using Python. If my understanding is incorrect, I welcome any corrections.

2. The model architecture and training strategy appear to be a combination of ChartMoE and ChartCoder. The former utilizes a Mixture-of-Experts  architecture with different routes for various chart understanding tasks, while the latter employs DeepSeek-Coder as the LLM to improve code generation performance. The overall model structure and training approach of CharLUMA are very similar to these two works, which may indicate a lack of novelty.

3. The evaluation data for Chart2NCode is sampled from the 176k dataset. This could result in the training and test data being drawn from the same distribution, which may inflate the reported performance metrics. I could not find any validation of this test set, analysis of the train/test distributions, or statistics on the quantity of different code types within the test set. This raises questions about the model's out-of-domain generalization capabilities.

**Questions:**

See Weakness

---

> ### Author Response · Authors · 2025-11-21
> **Author's Response to Reviewer A9S1**
>
> _**Weakness 1**: How the distribution mentioned in Section 6.2 (76.6% Python, 19.2% R, and 4.2% LaTeX) corresponds to the results in Figure 5 and how this ratio was sampled. Does this imply that the vast majority of the data is chart-to-Python and originates from ChartCoder?_
>
> Thank you for your questions. We would like clarify that our Chart2NCode dataset features the cross-language consistency, where each plotting script in the Python-R-Latex-Chart quadruple leads to the same chart image.
>
> First, the ratio “76.6% Python, 19.2% R, and 4.2% LaTeX” refers to **the provenance of language-agnostic metadata used to initialize the chart cases**. Each metadata record is rendered into plotting scripts in three languages through our automatic pipeline.
>
> Second, the predominance of metadata extracted from Python **reflects the bias of existing chart-to-code datasets towards one single language (Python)**. Models trained on such imbalanced data tend to exhibit weaker cross-language generalization, as observed in Figure 5, which motivates our balanced and cross-lingual alignment approach.
>
> You may refer to our response to Weakness 3 for the statistical breakdown of different code types.
>
> _**Weakness 2**: The model architecture and training strategy appear to be a combination of ChartMoE and ChartCoder, which may indicate a lack of novelty._
>
> We would like to highlight that **our model architecture features the low-rank subspace adaptation with language-guided routing mechanism**. Compared with independent experts in ChartMoE, we leverage a pool of low-rank subspaces to enable a shared representation space for multiple languages. This novel design enhances both cross-language generation and  in-language robustness.
>
> The projection layer in our work is also far more compact, as our low-rank adaptation design contains **only ~25% of the parameters of the mixture-of-MLP experts** used in ChartMoE.
>
> Finally, our two-stage training strategy follows the common practice in general MLLM training [1], to highlight the effectiveness of the proposed architectural design in our work.
>
> _**Weakness 3:** The training and test data is drawn from the same distribution, which raises questions about the model's out-of-domain generalization capabilities. I could not find any validation of this test set, analysis of the train/test distributions, or statistics on the quantity of different code types within the test set._
>
> We first would like to clarify that CharLuMA **exhibits strong generalisation capability on two out-of-domain benchmarks** ChartMimic and Plot2Code. CharLuMA-6.7B achieves 79.2 DS and 70.3 F1 on ChartMimic and 68.3 DS and 60.5 F1 on Plot2Code, representing the strongest performance among open-source models.
>
> Second, the data quality of Chart2NCode **has been validated through human evaluation in Section 3.2 and Appendix B.3**, which confirms its robust cross-language consistency.
>
> Third, we compute additional statistics on the dataset, including average code length and the distribution of chart types across the train and test splits. The average code lengths are 3998.5 and 4229.3 characters for the training and test sets, respectively. The dataset covers a broad range of chart types, including bar (18.8%), line (17.1%), scatter (15.4%), pie (12.4%), and radar (5.73%). More detailed statistics is incorporated into Appendix B.2 of our revised version.
>
> [1] Liu Haotian, et al. “Visual Instruction Tuning.” NeurIPS(2023).

---

> > ### Comment · Reviewer_A9S1 · 2025-11-28
> >
> > Thanks for the reponse. After reading the replies, I believe the original rating was reasonable.

---

### Author Response · Authors · 2025-11-30
**Revisions of Paper**

We extend our gratitude to all reviewers for their thorough review. We have updated the paper to comprehensively address the issues raised, with the revisions outlined as follows:

1. We conducted ablation studies on backbone choices in Section 6.1, testing a general-purpose MLLM (DeepSeek-LLM-7B) and an alternative vision encoder (CLIP-Large). As shown in Table 4, **our default configuration (DeepSeek-Coder-6.7B and SigLIP) yields superior performance** in both execution rate and image fidelity.
2. We broadened our baseline comparison to include more open-source MLLMs, such as the Qwen3-VL series and GLM-4.5v. Despite these additions, **CharLuMA-6.7B remains the top performer across all chart-to-code generation benchmarks.**
3. To mitigate code similarity inflation in models trained on in-distribution data, we replaced the code-side CrystalBLEU metric with image-side GPT-4o scoring for the Chart2NCode test set results in Section 5.2 and Table 1.
4. We added more detailed data statistics of Chart2NCode in Section 3.3 and Appendix B.2 including average code lengths, and proportions of examples over multiple chart types.
5. We investigated the impact of specific phrasing (e.g., "a STEM paper") in the evaluation prompt in Appendix C.3. The results indicate that variations in the prompt text have a negligible impact on overall model performance.
6. We updated Figure 3 with additional annotations and visual cues to improve the clarity of our data annotation pipeline.

---

### Note · Authors · 2026-01-03

I have read and agree with the venue's withdrawal policy on behalf of myself and my co-authors.